# CTSketch: Compositional Tensor Sketching for Scalable Neurosymbolic Learning

**Seewon Choi**[*]
University of Pennsylvania
seewon@seas.upenn.edu

**Alaia Solko-Breslin**[*]
University of Pennsylvania
alaia@seas.upenn.edu

**Rajeev Alur**
University of Pennsylvania
alur@seas.upenn.edu

**Eric Wong**
University of Pennsylvania
exwong@seas.upenn.edu

## Abstract

Many computational tasks benefit from being formulated as the composition of neural networks followed by a discrete symbolic program. The goal of neurosymbolic learning is to train the neural networks using end-to-end input-output labels of the composite. We introduce CTSketch, a novel, scalable neurosymbolic learning algorithm. CTSketch uses two techniques to improve the scalability of neurosymbolic inference: decompose the symbolic program into sub-programs and summarize each sub-program with a sketched tensor. This strategy allows us to approximate the output distribution of the program with simple tensor operations over the input distributions and the sketches. We provide theoretical insight into the maximum approximation error. Furthermore, we evaluate CTSketch on benchmarks from the neurosymbolic learning literature, including some designed for evaluating scalability. Our results show that CTSketch pushes neurosymbolic learning to new scales that were previously unattainable, with neural predictors obtaining high accuracy on tasks with one thousand inputs, despite supervision only on the final output. [2]

## 1 Introduction

Many computational tasks can be formulated as the composition of neural networks whose outputs are fed into a discrete program. Neurosymbolic learning aims to train the neural network using only end-to-end labels of the composite. Concretely, given a parameterized neural network $M_\theta$, a fixed symbolic program $c$, and training data $((x_1, \ldots, x_n), y)$, the goal is to minimize the loss $\mathcal{L}(c(M_\theta(x_1, \ldots, x_n)), y)$ to optimize $\theta$. The key challenge concerns computing the output distribution of $c$ with respect to its input distributions while ensuring the loss is fully differentiable.

One solution to this problem is to encode $c$ as a differentiable logic program, as Scallop [21], DeepProbLog [25], and DeepSoftLog [23] do. Scallop uses probabilistic Datalog to specify the symbolic program $c$ and can be configured with different provenance semirings, which determine how to aggregate proofs of each fact when evaluating a query. However, one limitation of many logic programming frameworks is that they often do not support external API calls to black-box modules within the symbolic program. This means they cannot use large language models (LLMs) to perform reasoning, which can be useful for certain tasks. For example, scene recognition has a natural decomposition of an object detector followed by a program that prompts an LLM to classify the scene based on the object predictions.

---

[*]Equal contribution.
[2]Code is available at https://github.com/alaiasolkobreslin/CTSketch

39th Conference on Neural Information Processing Systems (NeurIPS 2025).

Other neurosymbolic learning solutions instead treat $c$ as a black-box that can be encoded in any language and can include API calls. ISED [31] and IndeCateR [9] are techniques that fall into this category, as they both sample inputs to $c$ and compute the corresponding outputs to associate rewards with inputs. A-NeSI [32] is another black-box technique that uses a neural network to perform approximate inference over the *weighted model counting (WMC)* problem—the problem of computing the probability of each possible output by adding the probabilities of the corresponding input assignments–which is known to be computationally expensive [2]. While black-box learning algorithms can learn tasks that cannot be encoded as differentiable logic programs, they suffer from slow convergence and low accuracy on tasks with many inputs. This raises the question: can one design a more scalable solution by combining the strengths of white- and black-box techniques?

We propose CTSketch, a learning algorithm that encodes programs in tensors. In particular, the tensor *summary* $\phi$ captures the exact input-output relationship of $c$, i.e., $\phi[r] = c(r)$. Since the size of $\phi$ matches the input space of $c$, it may be computationally unaffordable to store $\phi$ if $c$ involves a large number of inputs. CTSketch uses two techniques to address scalability: decompose the symbolic program into *sub-programs* and sketch each sub-program summary. For white-box programs in which the internals are known, we can manually specify sub-programs that form a tree structure. In this architecture, inference proceeds through each sub-program by computing the product of its summary tensor and the input probabilities. The resulting distributions are passed on to the next sub-programs as inputs, and this process repeats until the last layer produces the final output distribution.

We use tensor sketching methods to reduce the size of sub-program summaries and increase the efficiency of inference. These techniques decompose a high-dimensional tensor into a product of low-rank tensors with low reconstruction error. Tensor sketching can be applied to any symbolic component, including black-box programs, regardless of the sub-program decomposition. With these sketches, we can perform inference via simple tensor operations and efficiently obtain the expected value of the output. From the reconstruction error of the sketching methods, we derive a bound on the maximum error of the approximation of the output distribution, providing insight into the performance of CTSketch.

In summary, the main contributions of this paper are as follows. First, we introduce program decomposition with tensor summaries as a way to scale neurosymbolic learning. Next, we introduce CTSketch, a scalable algorithm for learning neurosymbolic programs using composed tensor sketches. Then, we derive a bound on the maximum error of the approximation obtained by CTSketch. Finally, we conduct a thorough evaluation using state-of-the-art neurosymbolic frameworks against a diverse set of benchmarks. Our results demonstrate that CTSketch pushes the frontier of neurosymbolic learning, broadening its applicability to larger problems. In particular, it can solve the task of adding one thousand handwritten digits, which is a significantly larger problem than what prior works could solve, while remaining competitive with existing techniques on standard neurosymbolic benchmarks.

## 2  Overview

### 2.1  Problem Statement

For a task that involves unstructured inputs $(x_1, \ldots, x_n)$, the training pipeline consists of neural networks $M$ with parameters $\theta$, followed by a symbolic program $c : \mathcal{R} \to Y$ that computes a structured output $y$ given structured inputs $r_1, \ldots, r_n \in R_1 \times \cdots \times R_n = \mathcal{R}$. For each input $x_i$, the neural network outputs a probability distribution $p_i$ over discrete values (symbols) in $R_i$. We aim to estimate the distribution of the outputs of $c$ given the input distributions $p_i$, with the goal to optimize $\theta$ using only end-to-end training data $((x_1, \ldots, x_n), y)$, without intermediate supervision on $r$.

In order to predict the probability of each output $\hat{y} \in Y$ given input distributions $p_1, \ldots, p_n$, we aim to approximate the following WMC problem

$$\text{WMC}(\hat{y}|p_1, \ldots, p_n) = \sum_{r \in \mathcal{R}, c(r) = \hat{y}} \prod_{i=1}^{n} p_i(r_i) \qquad (2.1)$$

using tensors that *summarize* the program. We summarize the input-output pairs of the program using a single tensor $\phi : Y^{|R_1| \times \cdots \times |R_n|}$, which maps each input combination to its corresponding program

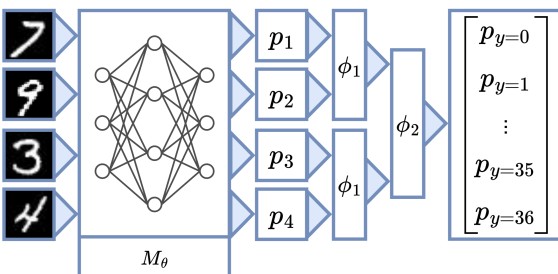

Figure 1: Program decomposition for $\text{sum}_4$. $\phi_1$ computes the sum of 2 digits, and $\phi_2$ computes the sum of the results from the first sums.

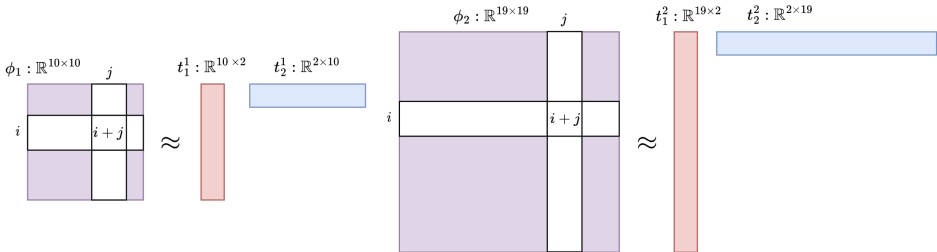

Figure 2: $\phi_1$ and $\phi_2$ for decomposed $\text{sum}_4$ without sketching (left of $\approx$) and with sketching (right of $\approx$). Sketching $\phi_i$ involves decomposing it as the product of rank-2 tensors $t_1^i$ and $t_2^i$.

output. It is also useful to represent $\phi$ using a one-hot tensor $\phi^{\text{OH}} : \{0,1\}^{|R_1| \times \cdots \times |R_n| \times |Y|}$:

$$\phi^{\text{OH}}[r_1, \ldots, r_n, \hat{y}] = \begin{cases} 1 & c(r_1, \ldots, r_n) = \hat{y} \\ 0 & \text{otherwise.} \end{cases}$$

During inference, after obtaining predicted distributions $p_i : [0,1]^{|R_i|}$ from the network, we can initialize $p$ to be their outer product, i.e., $p = \otimes_{i=1}^n p_i$, and compute the probability of each output $\hat{y} \in Y$, using Equation 2.1 with $\phi^{\text{OH}}$ as the indicator function:

$$p_{y=\hat{y}} = \sum_{r_1, \ldots, r_n} \phi^{\text{OH}}[r_1, \ldots, r_n, \hat{y}] \cdot p[r_1, \ldots, r_n]. \tag{2.2}$$

The problem with this approach is that the size of $\phi$ increases exponentially with the number of inputs, limiting its scalability. To address this challenge, we decompose $\phi$ and use a low-rank approximation of each component.

## 2.2 Program Decomposition and Sketching

To decompose $\phi$ for a given task, it is necessary to manually separate $\phi$ into sub-programs. For instance, the program that adds 4 handwritten digits can be decomposed as the sum of 2 sums (Fig. 1). The neural network predictions of the digits $p_i^1 = M_\theta(x_i)$ are used as inputs to the first 2-digit sum $\phi_1$. Inference through this first layer outputs distributions $p_j^2$, i.e., inputs to the second sum $\phi_2$.

While CTSketch can use any tensor sketching algorithm, we explain our method with tensor-train singular value decomposition (TT-SVD) [28]. The goal of TT-SVD is to find low-rank tensors, called *cores*, that reconstruct the original tensor with low error (Fig. 2). Given an approximation rank $\rho$, each core $t_i$ is built by reshaping the tensor into a matrix by flattening all except the $i$-th dimension, applying truncated SVD, and taking the top $\rho$ right singular vectors. With a large enough rank, TT-SVD can achieve an arbitrarily small error.

We use TT-SVD to sketch each sub-program summary $\phi_i : Y^{|R_1| \times \cdots \times |R_d|}$. Given $\rho$, TT-SVD outputs cores $(t_1^i, \ldots, t_d^i)$ where $t_1^i : \mathbb{R}^{1 \times |R_1| \times \rho}, t_d^i : \mathbb{R}^{\rho \times |R_d| \times 1}$, and all other $t_j^i : \mathbb{R}^{\rho \times |R_j| \times \rho}$. The full tensor reconstruction $\mathcal{T}_i$ is defined as

$$\mathcal{T}_i[l_1, \ldots, l_d] = \sum_{k_1, \ldots, k_{d-1}} \prod_{j=1}^d t_j^i[k_{j-1}, l_j, k_j] \tag{2.3}$$

where $k_0 = 1$ and $k_d = 1$. Note that we do not fully reconstruct $\mathcal{T}$ during inference in CTSketch, as this would not reduce memory overhead. Instead, we take the product of individual cores and input distributions to get an approximate output. We describe this process in more detail in §3.2.

This strategy differs from prior works on neurosymbolic scalability. The most closely related work is A-NeSI [32], which learns a neural network (prediction model) to predict the WMC result given probability distributions from the perception network (inference model). Therefore, the weights of both the inference and prediction models are learned during training. CTSketch circumvents this additional training complexity by requiring the task-specific program architecture to be user-defined and initialized by iterating through all (or some subset of) input-output pairs. These initialized tensors accurately capture the program semantics, and after sketching, the WMC approximation is still very accurate because sketching methods guarantee low error. This is in contrast to the DNN used by the inference model in A-NeSI, which can be trained to achieve high accuracy but lacks a theoretical guarantee on the error of the estimate.

## 3 CTSketch

### 3.1 Algorithm

When the symbolic program $c$ has many inputs, we manually decompose it into sub-programs that from a tree structure of $m$ layers. The leaves at layer 1 are neural network predictions $p_i^1 = M_\theta(x_i)$ and the root is the final output. The exact structure can be flexible, with one constraint to allow the computation to proceed sequentially from layer 1 to $m$: the sub-program at layer $i$ can only take outputs from the previous layers $p^{j<i}$ as inputs. The program does not need to be decomposed into a prefect tree as in Figure 1. For instance, it can contain bounded loops, which can be decomposed by unrolling into a sequence of repeated layers. For each sub-program, its input and output dimension need to be specified. For the $\text{sum}_4$ case, the input specifications would be $c_1^1 : (p_1^1, p_2^1), c_2^1 : (p_3^1, p_4^1)$ and $c_1^2 : (p_1^2, p_2^2)$.

We describe the steps of CTSketch using $\text{sum}_4$ (Fig. 1) and provide pseudocode in Appendix A. Prior to training, CTSketch initializes each $\phi_i$, using $c_i$, either by sampling a subset of the possible inputs to the sub-program or enumerating all combinations and computing the corresponding outputs. Note this does not require labeled data $(x, y)$ as we use structured input/output pairs $(r, y)$. After initializing each $\phi_i$, the algorithm obtains cores $(t_1^i, \ldots, t_d^i)$ by sketching $\phi_i$. The decomposition rank $\rho$ is a hyperparameter and can be chosen to be the full rank that equals the minimum product of the preceding or succeeding dimensions, $\min_i \left( \prod_{k=0}^{i-1} |R_k|, \prod_{k=i+1}^{d} |R_k| \right)$. In this case, the $i$-th core would exactly match the original tensor flattened into three dimensions, and the others would be identity matrices, ensuring a reconstruction error of zero.

After computing the sketches, for each training example $((x_1, \ldots, x_n), y)$ and its neural network predictions $(p_1^1, \ldots, p_n^1) = M_\theta(x_1, \ldots, x_n)$, CTSketch goes layer-by-layer through the program layers and computes the expected output for each sub-program.

### 3.2 Sketching Sub-Programs

We explain the inference steps by continuing with the $\text{sum}_4$ example. Assuming we use rank-2 TT-SVD decomposition, the program summary $\phi_1$ of the first layer decomposes into two rank-2 tensors $t_1^1$ and $t_2^1$, which are multiplied with network predictions $p_1^1$ and $p_2^1$ to produce $v \in \mathbb{R}$. Note that we can compute $v$ without explicitly reconstructing the full tensor (Equation 3.3).

$$v = \sum_a^{|R_1|} \sum_b^{|R_2|} \sum_x^2 p_1^1[a] p_2^1[b] t_1^1[a, x] t_2^1[x, b] \tag{3.1}$$

$$= \sum_x^2 \left( \sum_a^{|R_1|} p_1^1[a] t_1^1[a, x] \right) \left( \sum_b^{|R_2|} p_2^1[b] t_2^1[x, b] \right) \tag{3.2}$$

$$= (p_1^{1\top} t_1^1) \cdot (t_2^1 p_2^1) \tag{3.3}$$

We apply the RBF kernel and $L_1$ normalization to convert the value into a probability distribution $p_i^2 \in \mathbb{R}^{19}$, to be used as input in the following layers.

$$p_1^2[j] = \text{RBF}(v, j) = \exp\left(-\frac{1}{2\sigma^2}\|v - j\|_2\right)$$

Such transformation is not needed for the last layer, where the value can be directly compared with the ground truth for the loss computation (e.g., using $L_1$ loss). Hence, the final output space, computed by the last layer, can be infinite (e.g., floating-point outputs).

For programs with a finite output space, we can instead decompose the one-hot tensor summary $\phi_1^{\text{OH}}$ and obtain three rank-2 tensors: $t_1^1$, $t_2^1$ and $t_3^1$. Multiplying these with network predictions $p_1^1$ and $p_2^1$ results in a distribution $p_1^2 \in \mathbb{R}^{19}$. This is also computable without explicit reconstruction.

$$p_1^2[j] = \sum_a^{|R_1|} \sum_b^{|R_2|} \sum_x^2 \sum_y^2 p_1^1[a]p_2^1[b]t_1^1[a,x]t_2^1[x,b,y]t_3^1[y,j] \tag{3.4}$$

$$= \sum_y^2 t_3^1[y,j] \sum_x^2 \left(\sum_a^{|R_1|} p_1^1[a]t_1^1[a,x]\right) \left(\sum_b^{|R_2|} p_2^1[b]t_2^1[x,b,y]\right) \tag{3.5}$$

$$= \sum_y^2 t_3^1[y,j] \left(p_1^{1\top} t_1^1\right) \cdot \left(t_2^1[:,:,y]p_2^1\right) \tag{3.6}$$

The two distributions $p_1^2$ and $p_2^2$ are input to the second layer, and repeating this process produces the final output used for the loss computation. Note that we use the symbolic program $c$ at test time with the argmax inputs $r_i = \underset{j \in |R_i|}{\arg\max}(M_\theta(x_i))$ instead of the sketches.

## 3.3 Bound on Approximation Error

Suppose that we use TT-SVD to decompose $\phi_i$, and the tensor reconstructed from the cores is $\mathcal{T}_i$ (Equation 2.3). Let $\varepsilon_k$ be the *truncation error* of the decomposition: for the $k$th core, $\varepsilon_k$ is the Frobenius norm of the singular values discarded during the truncation step. With this, we can state the bound on the reconstruction error.

**Theorem 3.1** [28]. The reconstruction error of $\mathcal{T}$ satisfies

$$\|\phi_i - \mathcal{T}_i\|_F \leq \sqrt{\sum_{k=1}^{d-1} \varepsilon_k^2}. \tag{3.7}$$

In practice, the reconstruction error is very low, even for small values of approximation rank $\rho$. For example, in the $\text{sum}_n$ task, where the goal is to predict the sum of $n$ digits, we decompose the program into $\log_2(n)$ layers, with each sub-program computing the sum of its two inputs (Fig. 1). Even for $\text{sum}_{1024}$, where the summary tensor of the sub-program for the final layer, $\phi_{10}$, has shape $4609 \times 4609$, rank 2 is enough to give us a reconstruction error $\leq$ 1e-5 for all layers. With such a low reconstruction error, we can also guarantee that the product of each $\mathcal{T}_i$ with the corresponding input distributions will have a very low error. We derive an error bound of the output distribution if we were to do a full reconstruction of $\phi_i$.

**Theorem 3.2** (CTSketch error bound). For input distribution $p$, The error of the output distribution is bounded by

$$\|\phi_i^{\text{OH}}p - \mathcal{T}_i^{\text{OH}}p\|_2 \leq \sqrt{2}\|p\|_F \lfloor 4\|\phi_i - \mathcal{T}_i\|_F^2 \rfloor^{\frac{1}{2}}. \tag{3.8}$$

We provide a short proof, which uses the Cauchy-Schwartz inequality, in Appendix B. While using the full reconstruction is not exactly how CTSketch performs inference, it still gives us a good idea of how well the algorithm approximates the final distribution. CTSketch takes the product of each core, without reconstruction, with the respective input distribution and obtain an expected value of the output. This is then converted to a distribution with the RBF kernel and $L_1$ normalization.

# 4 Evaluation

In this section, we evaluate CTSketch and aim to answer the following research questions:

**RQ1:** Can CTSketch learn tasks with a large number of inputs that are not solvable by prior works?

**RQ2:** How does CTSketch perform on benchmarks that can be solved by existing techniques?

**RQ3:** How efficiently does CTSketch learn, i.e., how quickly does it converge compared to the baselines?

**RQ4:** How does the approximation rank of CTSketch influence its performance and training time?

## 4.1 Benchmark Tasks

We consider tasks from the neurosymbolic learning literature, including some designed for evaluating scalability.

**MNIST Sum.** We consider the problem of computing the sum of handwritten digits ($\text{sum}_n$) from the MNIST dataset [19]. We use sums of $n = 2^k$ digits, where $k \in \{2, 4, 6, 8, 10\}$. Each task uses a training set of 5K samples and a testing set of 1K samples, except $\text{sum}_{1024}$ where we use 4K training samples due to resource constraints.

**Multi-digit Addition.** We use the Multi-digit MNISTAdd task ($\text{add}_n$), originally proposed by [25], where the goal is to compute the sum of two $n$-digit numbers. We use $n \in \{1, 2, 4, 15, 100\}$ with a training set of $60,000/2n$ samples and a test set of $10,000/2n$ samples.

**Visual Sudoku.** We use the ViSudo-PC dataset [1] containing 200 4x4 and 2K 9x9 filled boards for training and testing. The goal of this task is to predict whether the input is a valid Sudoku board.

**Sudoku Solving.** The goal of this task is to solve a 9x9 Sudoku puzzle, where the board is given as a sequence of MNIST images with the digit 0 representing an empty cell. We use the SatNet [33] dataset with 9K training samples and 500 test samples and follow the same experimental setup as [4].

**HWF.** The Hand-Written Formula (HWF) task uses a dataset from [20] of 10K formulas of length 1–7 containing handwritten images of digits and operators. The dataset includes 1K length 1 formulas, 1K length 3 formulas, 2K length 5 formulas, and 6K length 7 formulas. The goal is to predict the result of the formula evaluation.

**Leaf Identification and Scene Classification.** We include two tasks from [31] which use GPT-4 to perform reasoning in the symbolic component for leaf identification and scene recognition, using datasets from [3] and [27] respectively. The goal is to identify a leaf species from the predicted features or classify a scene from the predicted objects.

## 4.2 Baselines and Experimental Setup

We choose several neurosymbolic techniques as baselines, with some specifically designed for scalability. We include Scallop [21], a framework that uses probabilistic Datalog to specify the symbolic component and has been shown to outperform DeepProbLog [25]. We also compare with DeepSoftLog (DSL) [23] for MNIST sum and multi-digit addition, but omit for other tasks since it cannot encode programs that use GPT-4 (leaf, scene), and requires customized solutions in which encoding complex reasoning is difficult (visudo, sudoku, hwf). We also consider IndeCateR [9], a scalable gradient estimator with lower variance than REINFORCE [34], ISED [31] that uses sampling and a summary logic program to obtain a custom loss, and A-NeSI [32], a framework for approximating WMC results by training a neural estimator.

We run all experiments on a machine with one 14-core Intel i9-10940X CPU, one NVIDIA RTX 3090 GPU, and 66 GB of RAM. We run each task with 10 random seeds and apply a timeout of 5 seconds per training example. We choose the number of epochs used for each technique based on when training saturates. For MNIST tasks, we use LeNet [19], a 2-layer CNN-based model, and we use a similar architecture for HWF and leaf. For the scene task, we use YOLOv8 [30] and a 3-layer CNN to detect objects. We also select the sample count, a key hyperparameter used in IndeCateR and ISED, based on the task, with the goal of balancing the training time while also giving the frameworks enough samples to learn. We describe the choices of other hyperparameters as well as the task decompositions in Appendices C.1 and C.2.

Table 1: Test accuracy results for $\text{sum}_n$ and $\text{add}_n$. The reference approximates the accuracy of a neural predictor for MNIST images with 0.99 accuracy using $0.99^n$ for $\text{sum}_n$ and $0.99^{2n}$ for $\text{add}_n$.

| | Accuracy (%) | | | | | | | |
|---|---|---|---|---|---|---|---|---|
| **Method** | $\text{sum}_{16}$ | $\text{sum}_{64}$ | $\text{sum}_{256}$ | $\text{sum}_{1024}$ | $\text{add}_2$ | $\text{add}_4$ | $\text{add}_{15}$ | $\text{add}_{100}$ |
| Scallop | 8.43 | TO | TO | TO | 95.3 | TO | TO | TO |
| DSL | 2.19 | 0.78 | TO | TO | 96.6 | **93.5** | **77.1** | **25.6** |
| IndeCateR | 83.01 | 44.43 | 0.51 | 0.60 | 93.3 | 89.0 | 69.6 | 6.4 |
| ISED | 73.50 | 1.50 | 0.64 | ERR | 93.1 | 89.71 | 0.0 | 0.0 |
| A-NeSI | 17.14 | 10.39 | 0.93 | 1.21 | 96.0 | 92.1 | 76.8 | ERR |
| CTSketch (Ours) | **83.84** | **47.14** | **7.76** | **2.73** | **96.7** | 92.51 | 74.8 | 23.5 |
| Reference | 85.15 | 52.56 | 7.63 | 0.003 | 96.06 | 92.27 | 73.97 | 13.40 |

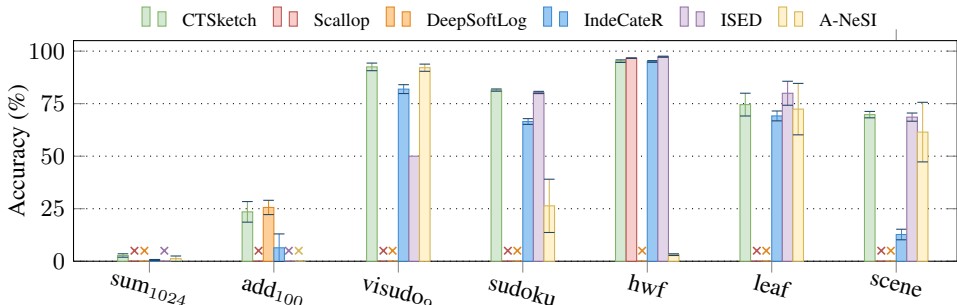

Figure 3: Test accuracy results for $\text{sum}_{1024}$, $\text{add}_{100}$, $\text{visudo}_9$, sudoku, HWF, leaf, and scene tasks. "X" indicates that there was a timeout, there was an error/overflow, or the given task was not able to be programmed in the framework. Error bars show standard deviations.

### 4.3 RQ1: Scalability

To answer **RQ1**, we increase the number of inputs to the $\text{sum}_n$ task up to 1024, which is orders of magnitude larger than previously studied input spaces. We report the $\text{sum}_n$ results for $n \in \{16, 64, 256, 1024\}$ in Table 1 and full results with standard deviations in Table 4 of Appendix C.3. The reference refers to the lower bound of accuracy when the neural predictor is supervised per-input to do individual digit recognition, computed as $0.99^n$ .

CTSketch is the best performer on all $n \geq 16$ and is the only method to consistently match the reference. All baseline methods fail to learn $\text{sum}_{256}$, whereas CTSketch also works for $\text{sum}_{1024}$. Even with very weak supervision on only the final sum of 1024 digits, CTSketch attains a per-digit accuracy of 93.69%. Although A-NeSI reaches similar accuracy, its per-digit accuracy stays at 17.92%, implying that the result is due to small dataset size and redundant use of training images.

While IndeCateR is overall the next best method, it faces a limit at 256 inputs and struggles even with 25,600 samples per example. ISED requires the most computational resources and exceeds the machine availability for $n \geq 64$ using the same sample count as IndeCateR. We reduce the sample count to 6,400 and 1,024 for $\text{sum}_{64}$ and $\text{sum}_{256}$, respectively, resulting in a significant performance drop. We report an error (ERR) for $\text{sum}_{1024}$ since even a sample count of one does not work.

Scallop, DSL, and A-NeSI are unable to learn efficiently from $\text{sum}_{16}$. Scallop times out with top-1 proofs for $n \geq 64$ since there are exponentially many input combinations to consider. By manually embedding the exact WMC inference, DSL attains the highest accuracy for $\text{sum}_4$. However, the exact inference times out for $\text{sum}_{16}$, and we use the alternative of training a surrogate neural network to predict the inference results. This neural embedding is unable to achieve high accuracy and eventually times out. A-NeSI also attempts to do neural approximation and similarly struggles to learn the inference model due to limited supervision.

### 4.4 RQ2: Accuracy

We compare the test accuracy of CTSketch with the baselines across 11 tasks from the neurosymbolic learning literature. We summarize the results of visual sudoku (visudo), sudoku, hwf, leaf, and scene

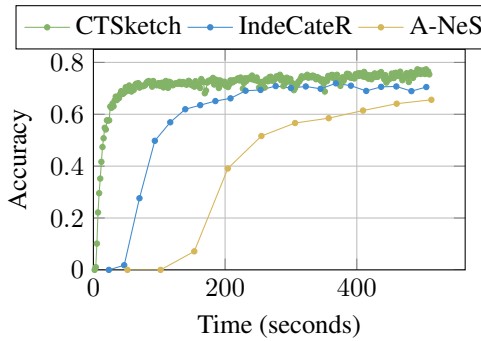 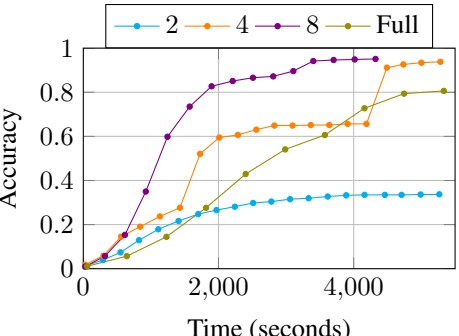

Figure 4: Accuracy vs. Time for $\text{add}_{15}$ for IndeCateR, A-NeSI, and CTSketch.

Figure 5: Accuracy vs. Time for different ranks $\rho \in \{2, 4, 8, \text{full}\}$.

in Figure 3, and we report the results for $\text{add}_n$ in Table 1. Full results with standard deviations can be found in Tables 3 and 5 (Appendix C.3). CTSketch is the highest performer on 4 tasks, including $\text{visudo}_9$, sudoku, and scene. Of all tasks in which CTSketch does not obtain the highest accuracy, the largest difference in accuracy is in $\text{visudo}_4$, where CTSketch comes within $2.55\%$ of A-NeSI. Furthermore, for every $\text{add}_n$ task, CTSketch comes within $5.42\%$ of the reference accuracy (of a 0.99 accuracy MNIST predictor) but often above the reference. These results demonstrate that even though CTSketch is designed for scalability, it can still solve a variety of classic neurosymbolic tasks.

No other baseline performs as consistently well as CTSketch across the benchmark tasks. Scallop times out of 5 of the tasks and cannot encode the leaf and scene tasks due to their use of GPT-4. The REINFORCE-based techniques, IndeCateR and ISED, do not scale as well as CTSketch because they cannot get a strong enough learning signal with the specified sample count. IndeCateR lags behind on sudoku and scene, and ISED struggles with scaling to $\text{visudo}_9$ and $\text{add}_n$ for large values of $n$. A-NeSI performs poorly on complex reasoning tasks like $\text{sudoku}_9$ and hwf and exhibits high variance. This is because A-NeSI trains an additional neural network to estimate the WMC result, and learns best when there is a stronger supervision than what is provided in some tasks. For example, A-NeSI scales well for $\text{add}_n$ where we provide per-digit supervision for the multi-digit output, but fails to scale for tasks like $\text{sum}_{1024}$ where there is weaker supervision on the final output.

## 4.5 RQ3: Computational Efficiency

We compare CTSketch to the baselines in terms of the test accuracy over training time on tasks: $\text{add}_{15}$ and $\text{add}_{100}$, and present the results in Figure 4 and Figure 6 (Appendix C.3). While the per-epoch accuracy improvement for CTSketch is not always the largest, CTSketch learns far faster than the baselines because of how efficiently it performs inference: the average epoch times for $\text{add}_{15}$ and $\text{add}_{100}$ are 1.70 and 0.92 seconds, respectively. The next fastest methods for $\text{add}_{15}$ were IndeCateR, taking 23.07 seconds, and A-NeSI, taking 52.72 seconds per epoch on average. While DSL shows comparable accuracy to CTSketch on these tasks, its learning is prohibitively slow, taking over 20 minutes per epoch. For some tasks, such as $\text{add}_{100}$, CTSketch converges before DSL even completes one training epoch. We can attribute this difference to the efficiency of inference. While other techniques require training a prediction network (as in A-NeSI) or learning a neural embedding (as in DSL), CTSketch requires only a short sequence of tensor operations to perform inference.

In our experiments, the overhead requirement of initializing and sketching sub-program summaries is very low, taking less than one minute across all tasks. This demonstrates that the overhead of sketching is negligible when considering the performance improvement it provides, allowing for extremely efficient inference and fast convergence.

## 4.6 RQ4: Sketching Rank

We study how the rank of sketching affects the accuracy and training time with the HWF task. We vary the rank $\rho \in \{2, 4, 8, \text{full}\}$ for sketching the largest tensor, which is of size $14^7$. We plot accuracy against time in Figure 5, where we mark the test accuracy for every 10 epochs during the first 1.5 hours of training. The comparison between the full rank and the others demonstrates the advantage of

sketching: when an appropriate rank is used, CTSketch can converge much faster without sacrificing accuracy. While the training time for a single epoch is similar across the lower ranks, it takes about twice the time using full rank. Consequently, while the rank-8 tensor converges to 95% accuracy in 70 minutes, the full rank tensor is still around 80% after 90 minutes.

The learning curve for rank 2 implies that the rank should be sufficiently large, or the neural network will fail to learn the optimal weights even with more training. We see time and space trade-offs in ranks 4 and 8. Since the number of entries for the sketched tensor is $(5\rho^2 + 2\rho) \times 14$, rank 8 requires 3.82 times more memory, but takes 30% less time to reach 90% accuracy. This demonstrates that training is not particularly sensitive to the choice of the rank, and can be chosen flexibly depending on the available resources. It is useful to consider both the reconstruction error $||\mathcal{T} - \phi||_F$ and the maximum difference $\text{MAX}(|\mathcal{T} - \phi|)$ when deciding the rank. For example, the maximum error is above 60 for the rank-4 approximation. Still, the model manages to learn as the reconstruction error is around 0.07, which corresponds to all entries in the tensor only differing by 1e-5.

## 5    Limitations and Future Work

The primary limitation of CTSketch lies in requiring manual decomposition of the program component to scale. When the full program summary tensor is computationally affordable, CTSketch is always applicable. However, when the input and output spaces are large, program decomposition is needed to fit program summaries in memory (Appendix D.1). The decomposition must be specified by the user, which may be difficult or impossible for complex tasks. In this sense, CTSketch can be viewed as a restrictive "language" in which not all tasks can be encoded. This motivates future work on automating the decomposition, possibly using program synthesis techniques.

Furthermore, it would be interesting to explore different tensor sketching methods and the trade-offs they provide, extending upon the initial results in Appendix D.2, where we compare TT-SVD against CP [13], Tucker [16], and Tensor Ring [37] decomposition methods. When using these methods, we initialize the tensor summary by enumerating all possible input combinations upfront, before sketching. If we instead use a streaming tensor sketching method, we would iteratively refine the sketches with a subset of input-output pairs at each iteration. A small amount of time overhead to update the sketches would result in a significant reduction in memory.

## 6    Related Work

**Neurosymbolic WMC techniques.** These frameworks provide techniques for computing the exact or approximate WMC result for a given input distribution and a program. [36] proposed a semantic loss function for learning with symbolic knowledge, which measures how well neural network outputs satisfy a given constraint using sampling for the WMC approximation. DeepProbLog [25] performs the exact WMC computation during inference, making it prohibitively expensive to use for complex tasks. Scallop [21] offers a more scalable solution by using provenance semirings that determine which proof terms to drop in the WMC approximation. Other solutions focus on treating the symbolic component as a black-box that can be written in any language; ISED [31] uses a sampling procedure, and A-NeSI [32] trains an additional neural network, to approximate the WMC result. While CTSketch and these methods both learn with a WMC estimate, CTSketch takes a different approach by performing inference using sketched tensor summaries. Inference in CTSketch does not aggregate proofs through a logic program (as in Scallop and DeepProbLog), or requires weight optimization of the approximation model (as in A-NeSI), resulting in more efficient learning.

**Other logic programming frameworks.** Other logic programming neurosymbolic techniques solve the learning problem without any exact or approximate weighted model counting during inference. [26] proposed extending DeepProbLog with an approximate inference technique, called DPLA*, which involves an A*-like search to obtain a small set of best proofs for a given query. DeepStochLog [35] uses stochastic (rather than probabilistic) logic programming to encode neurosymbolic programs and uses SLD resolution to get the probability of deriving a certain goal. On the other hand, DeepSoftLog [23] uses soft-unification to integrate embeddings into probabilistic logic programming. DeepSeaProbLog [11] also extends DeepProbLog by offering support for discrete-continuous random variables and uses weighted model integration, which can handle infinite sample spaces unlike weighted model counting. Abductive learning techniques [5, 14], which often use Prolog for the

symbolic component, offer an alternative solution that learns by by abducing pseudo-labels from ground-truth labels. These works use logic programming languages to encode the program component, unlike CTSketch which represents the program component with tensor summaries.

**Arithmetic and probabilistic circuits.** Arithmetic and probabilistic circuits (ACs and PCs) enable probability distributions to be computed in a fully differentiable manner with a combination of sum and product nodes [6, 7, 8]. This differentiability property makes them suitable for encoding the reasoning layer in neurosymbolic architectures. For example, a Semantic Probabilistic Layer has been proposed to integrate neural networks with logical constraints encoded as a PC and has been applied to tasks such as pathfinding [36]. However, one challenge with ACs and PCs in neurosymbolic learning is that it is difficult to run them on modern tensor accelerators. To solve this problem, PyJuice has been proposed as a general implementation design for running inference through PCs on the GPU [22]. Additionally knowledge layers (KLay) are a data structure for ACs that can be parallelized on GPUs [24]. Like ACs and PCs, decomposed program summaries in CTSketch also aim to make inference through the symbolic component fully differentiable. However, inference in CTSketch does not need additional data structures to improve efficiency since it uses only simple tensor operations that can already be performed on the GPU, and allows for tensor sketching to improve memory efficiency.

**Other neurosymbolic techniques.** There also exist neurosymbolic techniques that do not require the program component to be specified as a logic program. IndeCateR [9] offers a lower variance gradient estimator than REINFORCE [34] and can scale to complex black-box neurosymbolic programs. NASR [4] is a REINFORCE-style algorithm suited for fine-tuning models such as neural Sudoku solvers and scene graph predictors. ISED [31] also falls under this category of techniques that treat the neurosymbolic program as a black-box. While CTSketch is compatible with black-box programs, as in the leaf and scene tasks, the algorithm benefits from decomposing the symbolic component when there are many inputs. This allows CTSketch to scale far better than existing black-box frameworks, but it requires manual effort to decompose complex tasks.

Additionally, $PLIA_t$ [10] speeds up linear integer arithmetic by representing probabilistic integers as tensors and adapting FFT for sum operation, allowing efficient neurosymbolic inference. While $PLIA_t$ relies on tensor operations and decomposition of problems for scalability, similar to CTSketch, its applicability is limited to programs expressible using integer arithmetic. TerpreT [12] is a compositional rule framework that learns programs by inferring each instruction and its arguments sequentially. While program decomposition is central to both TerpreT and CTSketch, TerpreT is designed for inductive program synthesis whereas CTSketch is a neurosymbolic learning algorithm for training the neural component by backpropagating the gradient through the symbolic component.

## 7   Conclusion

We proposed CTSketch, a framework that uses decomposed programs to scale neurosymbolic learning. During inference, CTSketch uses sketched tensors, each representing the input-output summary of a sub-program, to efficiently approximate the output distribution of the symbolic component with simple tensor operations. We provide theoretical justification for our architecture and derive an error bound on the approximation. Our results show that CTSketch pushes the frontiers of neurosymbolic learning by solving previously unattainable tasks, such as adding one thousand handwritten digits.

## Acknowledgments and Disclosure of Funding

We thank the anonymous reviewers for the useful feedback. This research was supported by ARPA-H grant D24AC00253-00 and NSF award CCF 2313010.

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

# A Pseudocode

Without loss of generality, we assume in the pseudocode that there is only one program in each layer to simplify the notation. For example, we decompose the $sum_4$ program to consist of three layers, each with sub-programs $c_1$, $c_2$, and $c_3$, instead of two layers as in Figure 1.

---

**Algorithm 1** CTSketch training algorithm

---

**Require:** Symbolic programs $c_1, \ldots, c_m$ where $c_i$ takes $d_i$ inputs and outputs $y \in |Y_i|$, neural network $M_\theta$, loss function $\mathcal{L}$, and training dataset $\mathcal{D}$.

  **function** TRAIN($\mathcal{D}$)
    **for** $i = 1, \ldots, m$ **do**
      $t_1^i, \ldots, t_{d_i}^i \leftarrow$ INITIALIZE($c_i$)
    **end for**
    **for** $((x_1, \ldots, x_n), y) \in \mathcal{D}$ **do**
      $(p_1^1, \ldots, p_n^1) \leftarrow M_\theta(x_1, \ldots, x_n)$
      **for** $i = 1, \ldots, m$ **do**
        $(p_1, \ldots, p_{d_i}) \leftarrow$ INPUTS($c_i$)
        $v_i = \sum_x^{|p_1|} t_1^i[x] p_1[x]$
        **for** $k = 2, \ldots, d_i$ **do**
          $v_i = \sum_a^\rho \sum_x^{|p_k|} v_i[a] t_k^i[a, x] p_k[x]$
        **end for**
        **for** $j = 1, \ldots, |Y_i|$ **do**
          $p^{i+1}[j] \leftarrow$ RBF($v_i, j$)
        **end for**
        $p^{i+1} \leftarrow$ norm($p^{i+1}$)
      **end for**
      $l \leftarrow l + \mathcal{L}(v_m, y)$
    **end for**
    optimize $l$ and update $\theta$
  **end function**

---

# B Approximation Error Proof

Suppose that we sketch $\phi_i$ with TT-SVD, i.e., $(t_1^i, \ldots, t_d^i) \leftarrow$ SKETCH($\phi_i$), where the sketching rank is $\rho$. Let $\mathcal{T}_i$ be the reconstruction of $\phi_i$ using the sketches.

**Theorem 3.2** (CTSketch error bound). The error of the output distribution satisfies

$$\|\phi_i^{\text{OH}} p - \mathcal{T}_i^{\text{OH}} p\|_2 \leq \sqrt{2} \|p\|_F \lfloor 4 \|\phi_i - \mathcal{T}_i\|_F^2 \rfloor^{\frac{1}{2}}. \tag{B.1}$$

*Proof.* With some rearranging and by definition of the L2 norm, we have

$$\|\phi_i^{\text{OH}} p - \mathcal{T}_i^{\text{OH}} p\|_2 = \|(\phi_i^{\text{OH}} - \mathcal{T}_i^{\text{OH}}) p\|_2 \tag{B.2}$$

$$= \sqrt{\sum_{\hat{y} \in Y} \langle (\phi_i^{\text{OH}} - \mathcal{T}_i^{\text{OH}})[:, \ldots, :, \hat{y}], p \rangle_F^2}. \tag{B.3}$$

Applying the Cauchy-Schwartz inequality to the inner product, we obtain

$$\leq \sqrt{\sum_{\hat{y} \in Y} \|(\phi_i^{\text{OH}} - \mathcal{T}_i^{\text{OH}})[:, \ldots, :, \hat{y}]\|_F^2 \|p\|_F^2} \tag{B.4}$$

$$= \|p\|_F \sqrt{\sum_{\hat{y} \in Y} \|(\phi_i^{\text{OH}} - \mathcal{T}_i^{\text{OH}})[:, \ldots, :, \hat{y}]\|_F^2} \tag{B.5}$$

$$= \|p\|_F \sqrt{\sum_{\hat{y} \in Y} \sum_{r_1, \ldots, r_d} (\phi_i^{\text{OH}} - \mathcal{T}_i^{\text{OH}})[r_1, \ldots, r_d, \hat{y}]^2}. \tag{B.6}$$

Note that if $|\phi_i[r_1, \ldots, r_d] - \mathcal{T}_i[r_1, \ldots, r_d]| \geq 0.5$, then the one-hot encodings $\phi_i^{\text{OH}}[r_1, \ldots, r_d]$ differ in exactly two positions, which results in a squared difference of 2. Pulling this factor out, we obtain the desired result.

$$= \|p\|_F \sqrt{2 \sum_{r_1, \ldots, r_d} \mathbb{I}(|\phi_i[r_1, \ldots, r_d] - \mathcal{T}_i[r_1, \ldots, r_d]| \geq 0.5)} \tag{B.7}$$

By rearranging the definition of $\|\phi_i - \mathcal{T}_i\|$, the maximum, i.e. worst case, number of $r_1, \ldots, r_d$ such that $|\phi_i[r_1, \ldots, r_d] - \mathcal{T}_i[r_1, \ldots, r_d]| \geq 0.5$ is $\lfloor \frac{\|\phi_i - \mathcal{T}_i\|_F^2}{0.5^2} \rfloor = \lfloor 4\|\phi_i - \mathcal{T}_i\|_F^2 \rfloor$.

$$\leq \|p\|_F \sqrt{2 * \lfloor 4\|\phi_i - \mathcal{T}_i\|_F^2 \rfloor} \tag{B.8}$$

$$= \sqrt{2}\, \|p\|_F \lfloor 4\|\phi_i - \mathcal{T}_i\|_F^2 \rfloor^{\frac{1}{2}} \tag{B.9}$$

$\square$

## C    Experiment Details and Complete Results

### C.1    Experimental Setup and Hyperparameters

Unless stated otherwise, we keep the optimizer, training epochs, and batch size consistent across methods, and use the best learning rate among {1e-3, 5e-4, 2e-4, 1e-4, 5e-5}. For neural-GPT experiments, leaf classification and scene recognition, we copy the model, prompt and configuration from the original paper [31] where the tasks were introduced. The hyperparameters used for CTSketch are summarized in Table 2.

Table 2: Hyperparameters used in CTSketch for the benchmark tasks

| hyperparameter | visudo | sudoku | hwf | leaf | scene | $\text{sum}_n$ | $\text{add}_n$ |
|---|---|---|---|---|---|---|---|
| optimizer | Adam | AdamW | Adam | Adam | Adam | Adam | Adam |
| learning rate | 1e-3 / 5e-4 | 5e-6 | 1e-4 | 1e-4 | 5e-4 | 1e-3 / 5e-4 | 1e-3 |
| training epochs | 1000 / 5000 | 10 | 150 | 100 | 50 | 100 / 150 | 100 / 300 |
| batch size | 20 | 256 | 16 | 16 | 16 | 16 | 64 |
| sketching rank | full | 2 | 8 | full | full | 2 | full |

For the baseline methods Scallop, IndeCateR, ISED, A-NeSI, and DeepSoftLog we use:

**Visual Sudoku.** We use top-3 semiring for Scallop and learning rate 2e-4. We use $20 \times N \times N^2$ samples for IndeCateR, and learning rates 2e-4 and 5e-5 for 4x4 and 9x9 board, respectively. We use the same sample count for ISED and learning rates 1e-3 and 5e-4. For A-NeSI, we copy the hyperparameters used in their paper for this task.

**MNIST Sum.** For Scallop, we again use top-3 proofs and 1e-4 learning rate. For IndeCateR and ISED, we use a sample count of $100 \times n$. For IndeCateR, we run for 300 epochs, using batch size 10 learning rates 1e-3 and 5e-4. We use learning rates 1e-4 and 5e-5 for ISED, and reduce the sample count to 6400 and 1024 for $\text{sum}_{64}$ and $\text{sum}_{256}$. For DSL and A-NeSI, we copy the parameters used for MNIST-Add tasks in their papers. In the case of DSL, we use the neural embeddings for $\text{sum}_{16}$ with learning rate 1e-3 and 1e-5 weight decay. For A-NeSI, we decrease the learning rate to 5e-4, 2e-4, and 1e-4 for $n = 64, 256, 1024$. Due to memory constraints, we also reduce the embedding size from 800 to 400 for $n = 1024$.

**Multi-digit Addition.** For Scallop, we again use top-3 proofs and 1e-3 learning rate, and we use 30 epochs. For IndeCateR, we use a sample count of 10 and a 1e-3 learning rate. We train for 15 epochs for all values of $n$ except 100, where we use 45 epochs. These are the points when training saturates and accuracy starts to drop slightly with additional training. For ISED, we use a sample count of 100 and a 1e-4 learning rate. We train for 10 epochs because this is when training saturates. For A-NeSI and DeepSoftLog, we use the same hyperparameters used in their evaluation that have been optimized for multi-digit addition.

## C.2 Task Decomposition

**MNIST Sum.** These tasks involve adding $n$ digits, where $n$ is some power of 2. As a result, we structure the decomposition as a tree with $\log_2(n)$ layers. Each $\phi_i$ takes two inputs, each being integers between 0 and $9 * 2^i + 1$, and the result is the sum of the two inputs. Therefore, the shape of $\phi_i$ is $(9 * 2^i + 1) \times (9 * 2^i + 1)$, and its entries are the sum of the indices. Concretely, $\phi_i[a, b] = a + b$.

**Multi-digit Addition.** These tasks involve adding two $n$-digit numbers. We structure the decomposition as a chain of adds and carries. $\phi_1$ is the same as the first layer in MNIST sum, which takes 2 digits as inputs, and the possible outputs are $0 - 18$. For each place in the $n$-digit numbers, we add the two digits using $\phi_1$ in the first layer. For each of the $n - 1$ layers after this, we define $\phi_i$ as the carry sum, taking two inputs: the result of the current place sum and the result of the previous place's carry sum. The idea is that if the previous carry sum results in a 2-digit number, we carry over a 1, and the current place sum predictions get incremented by 1. This means each $\phi_i$ is a $19 \times 20$ tensor, with the first 10 column entries equal to the row index, and the last 10 column entries equal to one plus the row index.

**Visual Sudoku.** We decompose the task of determining whether the input is a valid sudoku board into pairwise comparisons of the cells. The cells in the same row, column, or block need to have distinct values, which leads to a total of $n = 56$ and 810 comparisons for $4 \times 4$ and $9 \times 9$ sudoku boards, respectively. Hence, the program consists of a single layer with $n$ instances of $\phi : N \times N$ such that $\phi[i, j] = i == j$, where $N \in \{4, 9\}$.

**Sudoku Solving.** For a single row, column, or block in a Sudoku board, if eight cells are filled with distinct values, the remaining one cell is uniquely defined as some value $v \in [1, 9]$. Using this idea, we decompose the Sudoku solving task into multiple instances of a program $c$ that computes the value $v$ given eight other values in the row, column, or block. We iterate through each cell, resulting in a total of $3 \times 81$ instances. We allow the inputs to be 0, meaning that it is unfilled and could be any value $n \in [1, 9]$. Note that the program is no longer a one-to-one mapping between the inputs and the output. For instance, if the eight inputs are all zeros, the output can be between 1 and 9. To encode one-to-many relationship we use $\phi^{OH} : 10 \times \cdots \times 9$, such that $\phi^{OH}[n1, \ldots, n8, v + 1] = 1$ for all $v$ satisfying the constraint. The case where $(n1, \ldots, n8)$ contains redundant values is naturally ignored as $\phi^{OH}[n1, \ldots, n8, :]$ becomes an all-zero vector.

**Handwritten Formula.** The HWF architecture uses only one layer, but we initialize 4 different $\phi_i$ tensors corresponding to each possible formula length (1, 3, 5, and 7). Since there are 14 possible symbols (10 digits and 4 arithmetic operators), $\phi_i$ has the same number of dimensions as the corresponding formula length, and each dimension has size 14. Each entry represents the real-valued output of the evaluated formula, and invalid formulas correspond to an output of 0.

**Leaf Identification and Scene Classification.** For these tasks that use GPT-4, the model internals are unknown, so we do not decompose them and instead initialize a single tensor summarizing the program. For the leaf task, there are 3 features (margin, texture, and shape) taking on 3, 3, and 6 possible values, respectively. The output values are one of the 11 possible leaf species. For the scene task, the object detector YOLOv8 returns a maximum of 10 objects per image, where each can be one of the 45 classes. The output values are one of the 9 possible room types.

## C.3 Full Experimental Results

We report full experimental results with 1-sigma standard deviation in Tables 3, 4, and 5, and plot test accuracy against training time on $\text{add}_{100}$ task in Figure 6. Note that the training for CTSketch converges before the first epoch of DeepSoftLog and ISED.

Table 3: Test accuracy on traditional neurosymbolic benchmarks.

| Method | Accuracy (%) | | | | | |
| --- | --- | --- | --- | --- | --- | --- |
| | $\text{visudo}_4$ | $\text{visudo}_9$ | sudoku | hwf | leaf | scene |
| Scallop | $84.92 \pm 3.11$ | TO | TO | $96.65 \pm 0.13$ | N/A | N/A |
| IndeCateR | $87.20 \pm 2.14$ | $81.92 \pm 2.11$ | $66.50 \pm 1.37$ | $95.08 \pm 0.41$ | $69.16 \pm 2.35$ | $12.72 \pm 2.51$ |
| ISED | $79.40 \pm 3.36$ | $50.0 \pm 0.0$ | $80.32 \pm 1.79$ | $\mathbf{97.34 \pm 0.26}$ | $\mathbf{79.95 \pm 5.71}$ | $68.59 \pm 1.95$ |
| A-NeSI | $\mathbf{91.90 \pm 1.90}$ | $92.11 \pm 1.71$ | $26.36 \pm 12.68$ | $3.13 \pm 0.41$ | $72.40 \pm 12.24$ | $61.46 \pm 14.18$ |
| CTSketch (Ours) | $89.35 \pm 2.45$ | $\mathbf{92.50 \pm 1.81}$ | $\mathbf{81.46 \pm 0.53}$ | $95.22 \pm 0.61$ | $74.55 \pm 5.42$ | $\mathbf{69.78 \pm 1.52}$ |

Table 4: Test accuracy results for $\text{sum}_n$. The reference approximates the accuracy of an MNIST predictor with 0.99 accuracy using $0.99^n$.

| | Accuracy (%) | | | | |
|---|---|---|---|---|---|
| **Method** | $\text{sum}_4$ | $\text{sum}_{16}$ | $\text{sum}_{64}$ | $\text{sum}_{256}$ | $\text{sum}_{1024}$ |
| Scallop | $88.90 \pm 10.78$ | $8.43 \pm 1.56$ | TO | TO | TO |
| DSL | $\mathbf{94.13 \pm 0.82}$ | $2.19 \pm 0.51$ | $0.78 \pm 0.32$ | TO | TO |
| IndeCateR | $92.55 \pm 0.87$ | $83.01 \pm 1.18$ | $44.43 \pm 10.19$ | $0.51 \pm 0.34$ | $0.60 \pm 0.23$ |
| ISED | $90.79 \pm 0.81$ | $73.50 \pm 2.73$ | $1.50 \pm 0.34$ | $0.64 \pm 0.26$ | ERR |
| A-NeSI | $93.53 \pm 0.35$ | $17.14 \pm 2.87$ | $10.39 \pm 0.92$ | $0.93 \pm 0.80$ | $1.21 \pm 1.28$ |
| CTSketch (Ours) | $92.17 \pm 0.43$ | $\mathbf{83.84 \pm 0.92}$ | $\mathbf{47.14 \pm 3.29}$ | $\mathbf{7.76 \pm 1.43}$ | $\mathbf{2.73 \pm 0.81}$ |
| Reference | 96.06 | 85.15 | 52.56 | 7.63 | 0.003 |

Table 5: Test accuracy results for $\text{add}_n$. The reference approximates the accuracy of an MNIST predictor with 0.99 accuracy using $0.99^{2n}$. For DSL and A-NeSI, we report the results from [23].

| | Accuracy (%) | | | | |
|---|---|---|---|---|---|
| **Method** | $\text{add}_1$ | $\text{add}_2$ | $\text{add}_4$ | $\text{add}_{15}$ | $\text{add}_{100}$ |
| Scallop | $96.9 \pm 0.5$ | $95.3 \pm 0.6$ | TO | TO | TO |
| DSL | $\mathbf{98.4 \pm 0.1}$ | $96.6 \pm 0.3$ | $\mathbf{93.5 \pm 0.6}$ | $\mathbf{77.1 \pm 1.6}$ | $\mathbf{25.6 \pm 3.4}$ |
| IndeCateR | $97.7 \pm 0.3$ | $93.3 \pm 1.3$ | $89.0 \pm 1.1$ | $69.6 \pm 2.1$ | $6.4 \pm 6.6$ |
| ISED | $91.4 \pm 9.1$ | $93.1 \pm 3.3$ | $89.7 \pm 1.1$ | $0.0 \pm 0.0$ | $0.0 \pm 0.0$ |
| A-NeSI | $97.4 \pm 0.3$ | $96.0 \pm 0.5$ | $92.1 \pm 1.1$ | $76.8 \pm 2.8$ | ERR |
| CTSketch (Ours) | $98.3 \pm 0.2$ | $\mathbf{96.7 \pm 0.4}$ | $92.5 \pm 1.1$ | $74.8 \pm 1.3$ | $23.5 \pm 4.9$ |
| Reference | 98.01 | 96.06 | 92.27 | 73.97 | 13.40 |

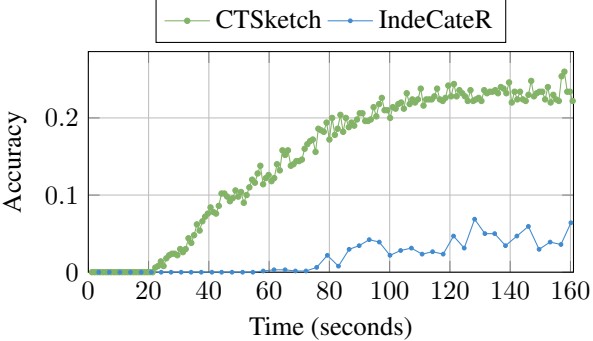

Figure 6: Accuracy vs. Time for $\text{add}_{100}$ for IndeCateR and CTSketch.

# D  Additional Ablation Studies

## D.1  Decomposition and Sketching

We compare the gains of program decomposition and tensor sketching for CTSketch using the $\text{sum}_{16}$ task. We vary the level of decomposition and compare accuracy, per-digit accuracy, and per-epoch training time for using full-rank tensors against rank-2 sketches in Table 6.

For the same decomposed program structure, we observe that sketching offers substantial memory savings, albeit with some loss in accuracy. When scaling to higher-dimensional input spaces, memory becomes a critical bottleneck – program decomposition alone is often insufficient, which highlights the necessity of tensor sketching for scaling. Within full-rank, the gains from greater degree of

decomposition are clear, both in terms of training time and memory efficiency. However, this trend reverses with sketches, due to the use of the RBF kernel at the output of each program layer.

Table 6: Ablation on gains of program decomposition and tensor sketching using $sum_{16}$. For *Dims*, 16 refers to no decomposition, whereas (8, 2) refers to 2-layer decomposition into two sum-8 at the first layer and a sum-2 at the second layer. *Method* refers to different inference procedures, where SKETCH uses rank-2 sketches, FULL uses one-hot program tensors $\phi^{OH}$ (Equation 3.6), and FULL + RBF use program tensors $\phi$ with RBF kernel (Equation 3.3). OOM refers to out-of-memory.

| Method | Dims | # Entries | Accuracy (%) | Digit Accuracy (%) | Time(s) |
|---|---|---|---|---|---|
| SKETCH | 16 | 600 | 83.52 | 98.86 | 1.72 |
| | 8, 2 | 572 | 83.88 | 98.90 | 2.11 |
| | 4, 4 | 588 | 83.25 | 98.85 | 2.57 |
| | 2, 2, 2, 2 | 556 | 82.51 | 98.78 | 3.88 |
| FULL | 16 | $1.45 \times 10^{18}$ | OOM | OOM | OOM |
| | 8, 2 | $7.30 \times 10^{9}$ | OOM | OOM | OOM |
| | 4, 4 | $2.72 \times 10^{8}$ | 85.59 | 99.01 | 17.66 |
| | 2, 2, 2, 2 | $8.88 \times 10^{5}$ | 85.80 | 99.04 | 2.0139 |
| FULL + RBF | 16 | $1.00 \times 10^{16}$ | OOM | OOM | OOM |
| | 8, 2 | $1.00 \times 10^{8}$ | 83.29 | 98.85 | 17.63 |
| | 4, 4 | $1.88 \times 10^{6}$ | 83.62 | 98.89 | 2.15 |
| | 2, 2, 2, 2 | $7.18 \times 10^{3}$ | 82.53 | 98.80 | 3.81 |

## D.2 Tensor Sketching Methods

We compare the performance of tensor sketching methods Tensor-Train, CP [13], Tucker [16], and Tensor Ring [37] on $sum_4$ and HWF in Table 7. While our method is not sensitive to the employed method, CP decomposition may be insufficient for complex input-output relationships, such as HWF, due to its restrictions to rank-1 tensors. Our experiments use Tensor-Train decomposition, which is not the best performer in terms of accuracy, but is competitive while requiring less space.

Table 7: Test accuracy on different tensor sketching methods on $sum_4$ and HWF tasks.

| | $sum_4$ | | | HWF | | |
|---|---|---|---|---|---|---|
| Method | Accuracy (%) | # Entries | Rank | Accuracy (%) | # Entries | Rank |
| Tensor Train | $92.17 \pm 0.43$ | 40 | 2 | $95.22 \pm 0.61$ | 1344 | 8 |
| CP | $92.33 \pm 0.59$ | 40 | 2 | $72.90 \pm 17.40$ | 1568 | 16 |
| Tucker | $92.31 \pm 0.62$ | 44 | 2 | $95.29 \pm 0.92$ | 16776 | 4 |
| Tensor Ring | $\mathbf{92.58 \pm 0.58}$ | 80 | 2 | $\mathbf{95.47 \pm 0.51}$ | 1568 | 8 |

# E License Information

To test the baselines, we used code from the official repositories of Scallop [15] (MIT), DeepSoftLog [23] (MIT), IndeCateR [9] (Apache 2.0), ISED [31] (MIT), and A-NeSI [32] (MIT). Additionally, for CTSketch, we used the implementation of TT-SVD from the python package tt-sketch [18] (CC BY-NC-ND 4.0) and cp, tucker, tensor ring decompositions from tensorly [17] (BSD 3-Clause).

We used several datasets in our evaluation, namely the Multi-illumination dataset [27] (CC BY 4.0), the handwritten formula dataset (CC BY-NC-SA 3.0) from NGS [20], and a subset of the leaf database [3] (CC BY 4.0). As part of the scene task experimental setup, we used YOLOv8 [30] (AGPL-3.0) and CLIP [29] (MIT).

