# OpenReview forum: "CTSketch: Compositional Tensor Sketching for Scalable Neurosymbolic Learning"
_NeurIPS.cc/2025/Conference — NeurIPS 2025 poster_

### Official Review · Reviewer_8NU1 · 2025-06-25

**Clarity:** 4
**Significance:** 3
**Originality:** 2
**Rating:** 5
**Confidence:** 4

**Summary:**

The work introduces CTSketch, a novel neurosymbolic learning algorithm that is designed to scale better.
The idea is to represent a logic program as a tensor, mapping each input combination to either 0 or 1, which still scales exponential in the number of variables. To scale, they approximate such a program by 1) decomposing it into several subprograms (currently limited to a user defined decomposition), which reduces the number of variables within a subprogram, and by 2) considering a low-rank approximation of those subprogram tensors. The empirical analysis demonstrates strong results.

**Questions:**

Not really questions, more feedback:

With regards to related work / future work, the authors may find interesting work on tensor decompositions for probabilistic inference in general. Some of these automatically identify decompositions based on the variable dependencies etc. E.g.,
- "Tensor Belief Propagation", ICML2017. Andrew Wrigley, Wee Sun Lee, Nan Ye
- "Parallel Weighted Model Counting with Tensor Networks. MCC2020. Jeffrey M. Dudek, Moshe Y. Vardi
- "Efficient Contraction of Large Tensor Networks for Weighted Model Counting through Graph Decompositions". 2019. Dudek et al.
- "Expressive power of tensor-network factorizations for probabilistic modelling". NeurIPS 2019. Glasser et al.
- "What is the Relationship between Tensor Factorizations and Circuits (and How Can We Exploit it)?" TMLR2025. Loconte et al.
- "Learning from Binary Multiway Data: Probabilistic Tensor Decomposition and its Statistical Optimality". JMLR2020. Wang et al.


Typo:
* "learning, Next, we ..."
* "Inference in CTSketch does not aggregate proofs", is an odd statement in a sense that the summations performed within CTSketch are aggregating in a similar fashion. I.e., you compute Eq 2.2, which is aggregating 'proofs'.

**Ethical Concerns:**

["NO or VERY MINOR ethics concerns only"]

**Final Justification:**

The authors have sufficiently addressed my questions. I remain positive about the work for the reasons outlined above.

**Limitations:**

yes

**Quality:**

3

**Strengths And Weaknesses:**

**Clarity**:
The primary ideas of the proposed work are easy to follow, and well-illustrated using Figures.

**Originality**:
The techniques proposed in this work exist; the novelty is their application in a neurosymbolic context. Having said that, the techniques only apply to the symbol-inference part, and are disconnected from the neural part of the pipeline. In that sense, it is not too surprising that this can be done. I was pleasantly surprised however by how well it appears to work in a neurosymbolic learning setting.

**Significance**:
The paper demonstrates strong empirical results, increasing the potential impact on the field. The fact that a user needs to manually specify a program decomposition limits the more general applicability of this work. However, I believe that existing related work may help here (see Questions section).

**Quality**:
The work is of good quality: the proof for the error bound is included in the appendix (but I did not verify it), the empirical experiments are in line with existing neurosymbolic papers, and the discussion of the empirical results are sufficiently informative/convincing. Something that could be of interest and that is currently missing is the impact of the program decomposition on the final results.

---

> ### Author Rebuttal · Authors · 2025-07-29
>
> > Originality: [...] Having said that, the techniques only apply to the symbol-inference part, and are disconnected from the neural part of the pipeline.
>
> The reason why we use the program summaries and tensor approximations is to connect the neural component with the symbolic program in a way that is end-to-end differentiable. In that sense, CTSketch is connected to the neural part of the pipeline.
>
> > Quality: [...] Something that could be of interest and that is currently missing is the impact of the program decomposition on the final results.
>
> |Method|Dims|# Entries|ACC (%)|DIGIT ACC (%)|TIME (s)
> |-|-|-|-|-|-
> |**SKETCH**|16|$600$|0.8352|0.9886|1.7154
> ||8, 2|$280, 292$ $(572)$|0.8388|0.9890|2.1066
> ||4, 4|$120, 468$ $(588)$|0.8325|0.9885|2.5705
> ||2,2,2,2|$40, 76, 148, 292$ $(556)$|0.8251|0.9878|3.8828
> |**FULL**|16|$10^{16}*145$|Out of Memory||
> ||8, 2|$10^8*{73}, 73^2*{145}$|Out of Memory||
> ||4, 4|$10^4*{37}, 37^4*{145}$ $(272123345)$|0.8559|0.9901|17.6586
> ||2,2,2,2|$10^2*{19}, 19^2*{37}, 37^2*{73}, 73^2*{145}$ $(887899)$|0.8580|0.9904|2.0139
> |**FULL+RBF**|16|$10^{16}$|Out of Memory||
> ||8, 2|$10^8, 73^2$|0.8329|0.9885|17.6282
> ||4, 4|$10^4, 37^4$ $(1884161)$|0.8362|0.9889 |2.1516
> ||2,2,2,2|$10^2, 19^2, 37^2, 73^2$ $(7159)$|0.8253|0.9880|3.8092
>
> We performed an ablation study on comparing the performance gains from program decomposition and tensor sketching with sum-16. We vary the level of decomposition and compare accuracy, per-digit accuracy, and per-epoch training time using full-rank vs rank-2 sketches. Here 16 refers to no decomposition, whereas (8, 2) refers to 2-layer decomposition into two sum-8 at the first layer and a sum-2 at the second layer.
>
> By comparing the full-rank (FULL) and rank-2 sketches configurations under the same decomposition structure, we observe that sketching offers substantial space savings, albeit with some loss in accuracy. As we scale to higher-dimensional input spaces, memory becomes a critical bottleneck—program decomposition alone is often insufficient. This highlights the necessity of tensor sketching for scaling.
>
> Within full-rank, the gains from greater degree of decomposition are clear, both in terms of training time and memory efficiency. However, this trend reverses with sketches, due to the use of the RBF kernel at the output of each program layer. For reference, we report results on using full-rank tensors with RBF kernels (FULL+RBF) which also show that  accuracy drops with more program layers
>
> >  With regards to related work / future work, the authors may find interesting work on tensor decompositions for probabilistic inference in general.
>
> Thank you for all these references. We will look into them for automating program decomposition.
>
>
> > "Inference in CTSketch does not aggregate proofs", is an odd statement in a sense that the summations performed within CTSketch are aggregating in a similar fashion. I.e., you compute Eq 2.2, which is aggregating 'proofs'.
>
> We agree that this statement is inaccurate. What we meant to say is that CTSketch does not aggregate proofs following the execution of a logic program, as is done in Scallop and DeepProbLog. CTSketch instead aggregates proofs through its tensor operations. We will clarify this point in the revised version of the paper.

---

> > ### Comment · Reviewer_8NU1 · 2025-08-05
> >
> > Thank you for the rebuttal, I have no further questions to discuss.

---

### Official Review · Reviewer_D6gf · 2025-07-02

**Clarity:** 3
**Significance:** 2
**Originality:** 2
**Rating:** 4
**Confidence:** 4

**Summary:**

The paper proposes a new method to perform scalable and approximate inference for neurosymbolic AI based on low-rank tensor decompositions of symbolic programs.

**Questions:**

1) Tensors (used in this paper) and arithmetic/probabilistic circuits (used in many other works) exhibit many similarities. It would be nice to see a discussion on these similarities in the context of the paper.

2) How was the full rank tensor implemented. Was it as dense multilinear function? The reason I am asking is because recent advances have shown that one can efficiently implement arithmetic circuits (ie sparse multilinear functions, ie sparse tensors) efficiently on the GPU and scale neurosymbolic learning without any approximations [2]. This might imply that scaling is mainly due to program decomposition and not low-rank tensor approxiamtions?

3) Is it not a bit of cheating to give direct supervision to the low-rank tensor decompositions of the sub-programs compared to what other methods are getting as training signal?

[2] "KLay: Accelerating Arithmetic Circuits for Neurosymbolic AI", Jaron Maene, Vincent Derkinderen, and Pedro Zuidberg Dos Martires, ICLR, 2025

**Ethical Concerns:**

["NO or VERY MINOR ethics concerns only"]

**Final Justification:**

See discussion with authors.

**Limitations:**

yes

**Paper Formatting Concerns:**

Looks good.

**Quality:**

3

**Strengths And Weaknesses:**

**Strengths**

The idea of using low-rank tensor decompositions to approximate symbolic programs in a differentiable fashion is novel.


**Weaknesses**

1) The paper claims that they are the first to use program decompositions to scale neurosymbolic learning. However, this is not the case. De Smet and Zuidberg Dos Martires [1] already used program decomposition to scale to similar problem sizes. Albeit for a restricted set of neurosymbolic problems. This reference is missing in the paper. Note, the techniques used in [1] was not explicitly called program decomposition.
2) The claim that logic programming languages do not support external API calls seems a bit strange. Logic programming is Turing-complete meaning they should probably be able to express an API call.

[1] "A Fast Convoluted Story: Scaling Probabilistic Inference for Integer Arithmetic", Lennert De Smet, and Pedro Zuidberg Dos Martires, NeurIPS, 2024

---

> ### Author Rebuttal · Authors · 2025-07-29
>
> > The paper claims that they are the first to use program decompositions to scale neurosymbolic learning. However, this is not the case.
>
> We will clarify in the revision that we are not the first to introduce general program decomposition. We do believe that CTSketch is the first neurosymbolic framework to use a decomposed program, represented as tensors, to improve the efficiency of inference. While a Scallop sum64 program would probably not benefit from decomposing it as 6 layers of sum programs (a timeout will still occur), CTSketch can drastically improve its efficiency when programs are decomposed.
>
> > De Smet and Zuidberg Dos Martires [1] already used program decomposition to scale to similar problem sizes. Albeit for a restricted set of neurosymbolic problems. This reference is missing in the paper.
>
> We will refer to PLIA in the Related Works section, explaining PLIA adapts FFT for efficient addition of probability distributions over integers, showing high speed-ups for programs expressible with integer arithmetics.
>
> > The claim that logic programming languages do not support external API calls seems a bit strange. Logic programming is Turing-complete meaning they should probably be able to express an API call.
>
> We made this claim in the context of differentiable logic programs not being compatible with calls to APIs. For example, one could not write a DeepProbLog program that calls GPT-4 as part of a neurosymbolic learning framework because it would not be end-to-end differentiable. However, some logic programming languages do offer foreign function interfaces like SWI-Prolog, which allows HTTP requests.
>
> > Tensors (used in this paper) and arithmetic/probabilistic circuits (used in many other works) exhibit many similarities. It would be nice to see a discussion on these similarities in the context of the paper.
>
> Indeed, there are several similarities between the tensors used in CTSketch and probabilistic circuits (PCs). PCs decompose their probability distribution computation with sum and product nodes. Product nodes in PCs compute the product of their inputs, where each input corresponds to a probability over a disjoint set of random variables. This is similar to how CTSketch takes the product of probabilities corresponding to every possible input combination. Sum nodes in PCs compute a probability distribution over the same set of random variables. This is similar to how for each possible output, CTSketch adds the probabilities of the input combinations that resulted in this output. Unlike PCs, CTSketch has the ability to compute an approximate output distribution extremely efficiently with tensor sketching. However, their similarities are important to highlight in the related work section, and we plan to do so in the next revision.
>
> > How was the full rank tensor implemented? Was it a dense multilinear function? The reason I am asking is because recent advances have shown that one can efficiently implement arithmetic circuits (ie sparse multilinear functions, ie sparse tensors) efficiently on the GPU and scale neurosymbolic learning without any approximations [2].
>
> Yes, the full rank tensor is implemented as a dense multilinear function. For a program $P: R_1 \times \dots \times R_n \rightarrow Y$, the full-rank tensor would be $\phi: R_1 \times \dots \times R_n$. This tensor can be represented as a arithmetic circuit consisting of three layers with $\sum_i |R_i|$ leaves, $\prod_i |R_i|$ nodes in layer 2, and $|Y|$ root nodes, similar to Figure 9 in [2]. KLay allows a time efficient neurosymbolic learning with this circuit, whereas CTSketch sketches $\phi$ for scalable inference..
>
> To clearly state how inference is done with the sketches, we further explain the computation of the equation between lines 135 and 136.
> \begin{align}
> v &= \sum_a^{|R_1|} \sum_b^{|R_2|} \sum_x^2 {p_1}[a] {p_2}[b] {t_1}[a, x] {t_2}[x, b] \\\\
> v &= \sum_x^2 \left( \sum_a^{|R_1|} {p_1}[a] {t_1}[a, x] \right)  \left(  \sum_b^{|R_2|} {p_2}[b]  {t_2}[x, b] \right) \\\\
> v &= ( p_1^\top t_1 ) \cdot ( t_2 p _2 )
> \end{align}
> We have vectors $p_1:|R_1|$ and $p_2:|R_2|$ that are probability distributions, and 2D matrices $t_1:|R_1|\times 2$ and $t_2:|R_2|\times 2$ (where 2 is the sketching rank). Instead of taking the outer product of $t_1 \times t_2$ and reconstructing the tensor, we can rearrange the order and do vector-matrix multiplication ($p_1$ and $t_1$, $p_2$ and $t_2$) and take the inner product of the two.
>
> > This might imply that scaling is mainly due to program decomposition and not low-rank tensor approxiamtions?
>
> |Method|Dims|# Entries|ACC (%)|DIGIT ACC (%)|TIME (s)
> |-|-|-|-|-|-
> |**SKETCH**|16|$600$|0.8352|0.9886|1.7154
> ||8, 2|$280, 292$ $(572)$|0.8388|0.9890|2.1066
> ||4, 4|$120, 468$ $(588)$|0.8325|0.9885|2.5705
> ||2,2,2,2|$40, 76, 148, 292$ $(556)$|0.8251|0.9878|3.8828
> |**FULL**|16|$10^{16}*145$|Out of Memory||
> ||8, 2|$10^8*{73}, 73^2*{145}$|Out of Memory||
> ||4, 4|$10^4*{37}, 37^4*{145}$ $(272123345)$|0.8559|0.9901|17.6586
> ||2,2,2,2|$10^2*{19}, 19^2*{37}, 37^2*{73}, 73^2*{145}$ $(887899)$|0.8580|0.9904|2.0139
> |**FULL+RBF**|16|$10^{16}$|Out of Memory||
> ||8, 2|$10^8, 73^2$|0.8329|0.9885|17.6282
> ||4, 4|$10^4, 37^4$ $(1884161)$|0.8362|0.9889 |2.1516
> ||2,2,2,2|$10^2, 19^2, 37^2, 73^2$ $(7159)$|0.8253|0.9880|3.8092
>
> We performed an ablation study on comparing the performance gains from program decomposition and tensor sketching with sum-16. We vary the level of decomposition and compare accuracy, per-digit accuracy, and per-epoch training time using full-rank vs rank-2 sketches. Here 16 refers to no decomposition, whereas (8, 2) refers to 2-layer decomposition into two sum-8 at the first layer and a sum-2 at the second layer.
>
> By comparing the full-rank (FULL) and rank-2 sketches configurations under the same decomposition structure, we observe that sketching offers substantial space savings, albeit with some loss in accuracy. As we scale to higher-dimensional input spaces, memory becomes a critical bottleneck—program decomposition alone is often insufficient. This highlights the necessity of tensor sketching for scaling.
>
> Within full-rank, the gains from greater degree of decomposition are clear, both in terms of training time and memory efficiency. However, this trend reverses with sketches, due to the use of the RBF kernel at the output of each program layer. For reference, we report results on using full-rank tensors with RBF kernels (FULL+RBF) which also show that  accuracy drops with more program layers.
>
> > Is it not a bit of cheating to give direct supervision to the low-rank tensor decompositions of the sub-programs compared to what other methods are getting as training signal?
>
> We do not provide any direct supervision using intermediate labels. We make the same assumptions as other neurosymbolic frameworks, which assume the domain of symbols (inputs to the program) is known. We use this assumption to randomly sample the symbols, prior to training and irrelevant to the actual training set labels, to initialize the program tensor.

---

> > ### Comment · Reviewer_D6gf · 2025-08-05
> >
> > I thank the authors for their detailed rebuttal.
> >
> > One point of contention: "one could not write a DeepProbLog program that calls GPT-4 as part of a neurosymbolic learning framework because it would not be end-to-end differentiable".
> >
> > This is not correct as DeepProblog and ProbLog have foreign function interfaces using Python.
> > https://dtai.cs.kuleuven.be/problog/tutorial/python/02-calling_python_from_problog.html
> >
> > As a matter of fact arithmetic operations implemented in any efficient logic programming language are implemented as foreign functions, cf. Chapter 8, of "The art of Prolog" Sterling&Shapiro.
> >
> >
> > Other than that I will retain my initial score.

---

### Official Review · Reviewer_o1QR · 2025-07-03

**Clarity:** 3
**Significance:** 2
**Originality:** 2
**Rating:** 4
**Confidence:** 3

**Summary:**

This paper targets at efficient computation of the output distributions of a program given the input variables' distribution. It assumes variables are discrete and the program/function can thus be modelled as a tensor mapping from choices of input variables' values to the output variable's value. They then apply tensor decomposition algorithms to approximate the tensor operations with low-rank and computationally efficient ones. The method also exploits the structure of programs. For example, summing n digits can be calculated as a tree of binary-sum sub-programs. They approximate each sub-program with tensor operations and then feed the outputted distributions to the next sub-program. The resulting method is differentiable and can thus support neuro-symbolic learning with gradient descent.

The method is evaluated in various classic neuro-symbolic tasks and shows better scalability for the sum_n program. It also performs competitively with previous methods in other tasks.

**Questions:**

* Are there more results regarding scalabilities e.g., for more programs and with more metrics? Can tensor operation approximation speed up general programs such as solving sudoku?
* Can tensor operation approximation be extended more realistic settings where the sub-program tensor summary is infeasible?

**Ethical Concerns:**

["NO or VERY MINOR ethics concerns only"]

**Final Justification:**

The paper is generally interesting while it would be better to have experiments on more complex programs.

**Quality:**

2

**Strengths And Weaknesses:**

This paper targets at an important problem: efficient approximations of the forward & backward computations of programs for neuro-symbolic learning. The method is interesting and incorporates the tensor decomposition techniques for approximation.

However, as discussed by the authors, tensor decomposition is only applicable when the full program summary tensor is computationally affordable. It is hard to apply those techniques in more realistic settings. Program decomposition itself is well-known and commonly used in the neuro-symbolic field such as in Terpret.

The experimental results are positive to some extent but are not strong enough to support claiming general improvements on scalabilities. The results about scalability are mainly for the sum_n program where program decomposition is easy. It is hard to tell if the scalability comes from program decomposition or tensor operation approximation. Just for sum_n, we could simply approximate its outputs' expectation as the summation of all input variables' expectations and get similar performance in my understanding.
The authors did not state the ranks to approximate each program in experiments as well. It would be cool to see faster sudoku approximations with e.g., partial input-output examples, although I am not sure if it's feasible.

---

> ### Author Rebuttal · Authors · 2025-07-29
>
> > [..] tensor decomposition is only applicable when the full program summary tensor is computationally affordable.
>
> In the paper, we construct the full program summary tensor before applying sketching, and mention streaming tensor sketching methods—where sketches are computed without explicitly forming the full tensor—as a direction for future work.
> However, in response to your comment, we conducted an additional experiment on the sum-16 task without program decomposition (results reported below). In this setting, the full summary tensor would have size $10^{16}$, making direct construction computationally infeasible. Instead, we employed a cross approximation technique to learn rank-2 sketches directly from a subset of input-output samples, without forming the full tensor. We used the implementation provided by the tntorch Python package. A more thorough evaluation of how cross approximation can be integrated into CTSketch is left for future work.
>
>
> > Program decomposition itself is well-known and commonly used in the neuro-symbolic field such as in Terpret.
>
> We should clarify that CTSketch isn’t the first work to propose program decomposition. TerpreT does have a compositional rule framework which guides their program synthesis. But TerpreT is a language designed for inductive program synthesis, and its programs can be used in the neurosymbolic learning setting. However, we believe that CTSketch is the first neurosymbolic learning framework to use a decomposed program, represented as tensors, to improve the efficiency of inference. While a Scallop sum64 program would probably not benefit from decomposing it as 6 layers of sum programs (a timeout will still occur), CTSketch can drastically improve its efficiency when programs are decomposed. We will clarify in the revision that we are introducing program decomposition with tensor summaries, not just general program decomposition.
>
> > The authors did not state the ranks to approximate each program in experiments as well.
>
> We report the ranks for each task in Table 2 (Appendix C) along with other hyperparameters.
>
> > It is hard to tell if the scalability comes from program decomposition or tensor operation approximation. [...] Are there more results regarding scalabilities e.g., for more programs and with more metrics?
>
> We performed an ablation study on comparing the performance gains from program decomposition and tensor sketching with sum-16. We vary the level of decomposition and compare accuracy, per-digit accuracy, and per-epoch training time using full-rank vs rank-2 sketches. Here 16 refers to no decomposition, whereas (8, 2) refers to 2-layer decomposition into two sum-8 at the first layer and a sum-2 at the second layer.
>
> |Method|Dims|# Entries|ACC (%)|DIGIT ACC (%)|TIME (s)
> |-|-|-|-|-|-
> |**SKETCH**|16|$600$|0.8352|0.9886|1.7154
> ||8, 2|$280, 292$ $(572)$|0.8388|0.9890|2.1066
> ||4, 4|$120, 468$ $(588)$|0.8325|0.9885|2.5705
> ||2,2,2,2|$40, 76, 148, 292$ $(556)$|0.8251|0.9878|3.8828
> |**FULL**|16|$10^{16}*{145}$|Out of Memory
> ||8, 2|$10^8*{73}, 73^2*{145}$|Out of Memory
> ||4, 4|$10^4*{37}, 37^4*{145}$ $(272123345)$|0.8559|0.9901|17.6586
> ||2,2,2,2|$10^2*{19}, 19^2*{37}, 37^2*{73}, 73^2*{145}$ $(887899)$|0.8580|0.9904|2.0139
> |**FULL+RBF**|16|$10^{16}$|Out of Memory
> ||8, 2|$10^8, 73^2$|0.8329|0.9885|17.6282
> ||4, 4|$10^4, 37^4$ $(1884161)$|0.8362|0.9889 |2.1516
> ||2,2,2,2|$10^2, 19^2, 37^2, 73^2$ $(7159)$|0.8253|0.9880|3.8092
>
> By comparing the full-rank (FULL) and rank-2 sketches configurations under the same decomposition structure, we observe that sketching offers substantial space savings, albeit with some loss in accuracy. As we scale to higher-dimensional input spaces, memory becomes a critical bottleneck—program decomposition alone is often insufficient. This highlights the necessity of tensor sketching for scaling.
>
> Within full-rank, the gains from greater degree of decomposition are clear, both in terms of training time and memory efficiency. However, this trend reverses with sketches, due to the use of the RBF kernel at the output of each program layer. For reference, we report results on using full-rank tensors with RBF kernels (FULL+RBF) which also show that  accuracy drops with more program layers. Sketching ranks are reported in Table 2 (Appendix C)  along with other hyperparameters.
>
> > Can tensor operation approximation speed up general programs such as solving sudoku?
>
> Encoding programs as tensors provides a faster way to do inference and gradient propagation  on general programs compared to other neurosymbolic frameworks. Tensor sketching provides additional benefit in terms of memory, in trade-off of performance. For high-dimensional inputs, smaller sized tensors allow for faster inference.
>
> > Can tensor operation approximation be extended to more realistic settings where the sub-program tensor summary is infeasible?
>
> The additional results reported above include solving sum-16 without program decomposition, for which using a full program tensor is computationally infeasible. Furthermore, the sum-8 sub-program tensor when decomposing into sum-8 and sum-2 also does not fit in memory. In both cases, sketching those tensors provides a solution.

---

> > ### Comment · Reviewer_o1QR · 2025-08-08
> >
> > I appreciate the authors’ efforts and think the new experiments regarding cross approximation techniques are interesting. I will remain supportive for acceptance and would encourage the authors to investigate more about general programs/tasks.

---

### Official Review · Reviewer_hw4L · 2025-07-03

**Clarity:** 2
**Significance:** 2
**Originality:** 3
**Rating:** 4
**Confidence:** 3

**Summary:**

In this paper, the authors propose CTSKetch, a framework to efficiently train neuro-symbolic models by using only end-to-end training data (i.e., without intermediate supervision on the representations extracted by the neural component). To achieve this, CTSKetch summarizes all possible relations between inputs and outputs (the task semantics) in a single tensor, mapping inputs (representations extracted from the neural component) to outputs (labels). To tackle the potential combinatorial explosion in size of this tensor, the authors propose two remedies: decomposition of the full tensorized program into composable sub-programs and low-rank decomposition of task-tensors (TT-SVD is used in the paper but could be any low-rank decomposition as suggested in Future Work). The authors provide an approximate error bound (valid for when TT-SVD is used as LoRa approximation) for the proposed CTSKetch. Finally, the authors present experimental results on different well-known benchmark in the neuro-symbolic community (MNIST sum/addition, visual sudoku, sudoku solving, HVF, leaf identification and scene classification), showing that 1) it overall scales better compared to other baselines (e.g., increasing the number of operands in the sum), 2) it achieves competitive results in terms of task accuracy, and 3) it converges faster compared to other baselines.

**Questions:**

Please see weaknesses.

**Ethical Concerns:**

["NO or VERY MINOR ethics concerns only"]

**Final Justification:**

The idea is original. The clarifications and additional experimental results provided during the rebuttal solved some of my initial concerns about the work. However, I still have concerns regarding the scalability of the proposed approach beyond the very simplistic settings investigated in the paper. I also hope that the authors, in case of acceptance, will effectively spend time increasing the quality of the write-up, which in the initial submission was not exceptional.

**Limitations:**

yes

**Quality:**

3

**Strengths And Weaknesses:**

**Strengths**

The paper was overall well-written, and I enjoyed reading it. The idea of using low-rank approximation to avoid the instantiation of the full task tensor at one time in memory is nice, and so is the perpendicular contribution of decomposing the full task into a composition of sub-tasks. Furthermore, the authors provide a sufficiently wide range of experiments on the CTSKetch, with some statistical significance obtained by using multiple (10) seeds.

**Weaknesses**

- Different aspects of this submission do not fully convince me. Firstly, the manual decomposition of tasks necessary for scaling (also mentioned by the authors themselves, which I appreciate) seems indeed a significant limitation for the applicability of the method outside of the simple domains investigated in this work. You mentioned that program synthesis could potentially come to the rescue in that case. Could you argue more about it?


- For my understanding, CTSKetch requires the explicit programming of the task tensor, which also sounds very limited in terms of scalability and applicability to broader domains. You mention in the paper that the construction of this vector could also be automated (using a subset of the training), but this would require labels for the intermediate representations, which would be against the initial proposition of working in an end-to-end setting.


- On another note, for my understanding, using low-rank approximation allows the avoidance of the instantiation of the full (sub-) task tensor in memory. However, this still needs to be computed fully (see Equation between Lines 135-136). So, if from a computation perspective the LoRa approximation only worsens the efficiency, where do the improvements shown in Section 4.5 come? Is it only the decomposition of the full task into multiple subtasks? I suggest that the authors elaborate more on this regard in the manuscript to increase its soundness, possibly including additional ablations regarding the effect of the single components (LoRa approximation, sub-task decomposition) on both accuracy and computational efficiency for at least some of the investigated tasks.


- More generally, I also question the generalizability of encoding the task semantics with a single input-output tensor, which is, by construction, only going to work with known input examples. For instance, a network trained on sum4 would not generalize to sum5 out of the box, requiring the manual extension of its entries to cover those examples which are not in $sum4 \cap sum5$.


- Regarding the writing, I think that the quality of the paper is acceptable, but some parts are not extremely clear (especially Sections 1 and 2) due to notation clutter or imprecise/verbose explanations. For instance, in Equation 2.1 what does p_i[r_i] refers to? Shouldn't it be sufficient to have \sum p_i? In my understanding, p_i is already the distribution associated with r_i. Or also, I didn’t find Figure 2 helpful right now, while, for example, populating the matrices with actual values and guiding a bit more the readers with some captioning could really increase the effectiveness and helpfulness of this figure. Also, why in Figure 1 is the fourth digit separated from the others? Also, in Line 147, shouldn’t it be “Approximate Error Bound”?

---

> ### Author Rebuttal · Authors · 2025-07-29
>
> >  You mentioned that program synthesis could potentially come to the rescue in that case. Could you argue more about it?
>
> As a starting point for synthesizing decomposed programs, we asked Gemini 2.5 Flash how it would decompose Sum16: “How would you decompose the program that adds 16 digits, into a tree-based structure, using sub-programs that take as few inputs as possible…” Its response was the same as what we came up with:
>
> **Depth:** 4 layers (excluding the initial input layer).
>
> **Components:**
> - Layer 0 (Inputs): 16 single-digit inputs.
> - Layer 1: 8 ADD sub-programs. Each takes 2 single-digit inputs.
> - Layer 2: 4 ADD sub-programs. Each takes 2 multi-digit inputs from Layer 1.
> - Layer 3: 2 ADD sub-programs. Each takes 2 multi-digit inputs from Layer 2.
> - Layer 4 (Root): 1 ADD sub-program. Takes 2 multi-digit inputs from Layer 3.
>
> After two steps of prompting, we were also able to get a pairwise comparison decomposed program for Visudo. We think that LLMs could be a first step towards automating the decomposition process.
>
> > Firstly, the manual decomposition of tasks necessary for scaling (also mentioned by the authors themselves, which I appreciate) seems indeed a significant limitation for the applicability of the method outside of the simple domains investigated in this work.
>
> We are also currently investigating CTSketch’s applicability to real-world domains such as predicting long-horizon sepsis from electronic health record (EHR) data. We are working on training a transformer to predict future EHR data from previous entries, and using a SOFA score program (which measures the severity of organ failure in a patient) with CTSketch to add an additional learning signal for predicting sepsis.
>
> > You mention in the paper that the construction of this vector could also be automated (using a subset of the training), but this would require labels for the intermediate representations, which would be against the initial proposition of working in an end-to-end setting.
>
> We did not mean to imply that the construction of the task tensor would use a subset of the training data. What we meant was that the task tensor can use a subset of all possible structured input combinations, which doesn’t require any labeled data. For example, to initialize a sum2 tensor, we would need to know that the possible values (symbols) for both inputs are integers between 0 and 9, and we could sample some subset of the possible symbol combinations (and their corresponding outputs) to populate the summary tensor.
>
> > On another note, for my understanding, using low-rank approximation allows the avoidance of the instantiation of the full (sub-) task tensor in memory. However, this still needs to be computed fully (see Equation between Lines 135-136). So, if from a computation perspective the LoRa approximation only worsens the efficiency, where do the improvements shown in Section 4.5 come?
>
> Due to properties of tensor products, we can compute the value of the Equation between lines 135-136 without reconstructing the tensor.
>
> \begin{align}
> v &= \sum_a^{|R_1|} \sum_b^{|R_2|} \sum_x^2 {p_1}[a] {p_2}[b] {t_1}[a, x] {t_2}[x, b] \\\\
> v &= \sum_x^2 \left( \sum_a^{|R_1|} {p_1}[a] {t_1}[a, x] \right)  \left(  \sum_b^{|R_2|} {p_2}[b]  {t_2}[x, b] \right) \\\\
> v &= ( p_1^\top t_1 ) \cdot ( t_2 p _2 )
> \end{align}
>
> $p_1:|R_1|$ and $p_2:|R_2|$ are vectors representing probability distributions, and $t_1:|R_1|\times 2$ and  $t_2:|R_2|\times 2$ are rank-2 sketches. By rearranging the equation, we see that, instead of taking the output product of $t_1 \times t_2$, we can obtain the result with  two vector-matrix multiplications and taking the inner product of the resulting vectors.
>
> > I suggest that the authors elaborate more on this regard in the manuscript to increase its soundness, possibly including additional ablations regarding the effect of the single components (LoRa approximation, sub-task decomposition) on both accuracy and computational efficiency for at least some of the investigated tasks.
>
> We performed an ablation study on comparing the performance gains from program decomposition and tensor sketching with sum-16. We vary the level of decomposition and compare accuracy, per-digit accuracy, and per-epoch training time using full-rank vs rank-2 sketches. Here 16 refers to no decomposition, whereas (8, 2) refers to 2-layer decomposition into two sum-8 at the first layer and a sum-2 at the second layer.
>
> |Method|Dims|# Entries|ACC (%)|DIGIT ACC (%)|TIME (s)
> |-|-|-|-|-|-
> |**SKETCH**|16|$600$|0.8352|0.9886|1.7154
> ||8, 2|$280, 292$ $(572)$|0.8388|0.9890|2.1066
> ||4, 4|$120, 468$ $(588)$|0.8325|0.9885|2.5705
> ||2,2,2,2|$40, 76, 148, 292$ $(556)$|0.8251|0.9878|3.8828
> |**FULL**|16|$10^{16}*{145}$|Out of Memory
> ||8, 2|$10^8*{73}, 73^2*{145}$|Out of Memory
> ||4, 4|$10^4*{37}, 37^4*{145}$ $(272123345)$|0.8559|0.9901|17.6586
> ||2,2,2,2|$10^2*{19}, 19^2*{37}, 37^2*{73}, 73^2*{145}$ $(887899)$|0.8580|0.9904|2.0139
> |**FULL+RBF**|16|$10^{16}$|Out of Memory
> ||8, 2|$10^8, 73^2$|0.8329|0.9885|17.6282
> ||4, 4|$10^4, 37^4$ $(1884161)$|0.8362|0.9889 |2.1516
> ||2,2,2,2|$10^2, 19^2, 37^2, 73^2$ $(7159)$|0.8253|0.9880|3.8092
>
> By comparing the full-rank (FULL) and rank-2 sketches configurations under the same decomposition structure, we observe that sketching offers substantial space savings, albeit with some loss in accuracy. As we scale to higher-dimensional input spaces, memory becomes a critical bottleneck—program decomposition alone is often insufficient. This highlights the necessity of tensor sketching for scaling.
>
> Within full-rank, the gains from greater degree of decomposition are clear, both in terms of training time and memory efficiency. However, this trend reverses with sketches, due to the use of the RBF kernel at the output of each program layer. For reference, we report results on using full-rank tensors with RBF kernels (FULL+RBF) which also show that  accuracy drops with more program layers.
>
> > More generally, I also question the generalizability of encoding the task semantics with a single input-output tensor, which is, by construction, only going to work with known input examples. For instance, a network trained on sum4 would not generalize to sum5 out of the box, requiring the manual extension of its entries to cover those examples which are not in sum4 \cap sum5
>
> We want to make it clear which components do and do not generalize to different tasks. It is true that a program summary tensor for sum4, representing the program’s discrete inputs and outputs, would not generalize to sum5. However, the same could be said for existing neurosymbolic approaches, i.e., a DeepProbLog program that sums 2 digits cannot be used to train sum4. However, the neural network used to train sum2 in CTSketch (with sketching) does generalize to sum4 at test time. Over 10 random seeds, we obtained an average of 92.5% accuracy and 98.1% digit accuracy on sum4, using a neural network trained on sum2.
>
> > Regarding the writing, I think that the quality of the paper is acceptable, but some parts are not extremely clear (especially Sections 1 and 2) due to notation clutter or imprecise/verbose explanations.
>
> Thank you for this feedback. To clarify, in Equation 2.1, we are considering a program with $n$ inputs, so we have probability distributions over each input $p_1$ through $p_n$. $r_i$ refers to the $i$th discrete input in the input combination $r_1, \dots, r_n$ that results in output $\hat{y}$. So $p_i[r_i]$ refers to the probability of this input value. Since this notation is confusing, we will revise it to $$\text{WMC}(\hat{y} \mid p_1, \dots, p_n) = \sum_{\mathbf{r} \in \mathcal{R},~ c(\mathbf{r}) = \hat{y}} \prod_{i=1}^n p_i(r_i)$$
>
> where $r=(r_1, \dots,r_n​) \in R_1 \times \dots \times R_n$. We hope that this will remove some ambiguity around indexing. For Figure 2, we thought that populating these matrices with values would make the figure look cluttered, but we can see why only explaining how they are populated could be unclear. In the revised version, we will add these numbers to the figure either in the main body or in the appendix. We can fix Figure 1 to make it clear that the 4th digit is also an input to the neural network, and we will change the section title to “Bound on Approximation Error”

---

> > ### Comment · Reviewer_hw4L · 2025-08-04
> >
> > Thanks for your detailed rebuttal and for the additional ablation study! I have additional comments on the author's response.
> >
> > - Being aware of the limitations that LLMs face, I am to some extent skeptical about their applicability to the program synthesis task, at least for tasks of complexity greater than Sum16 and Visudo. However, I am also aware that this goes beyond the scope of the paper, so I acknowledge that this does not represent a limitation of the current submission (but, rather, of the long-term applicability of the proposed technique).
> >
> > - Regarding the additional real-world experiment you mention on EHR, it is not exactly clear what the role of CTSketch is in this case. Would you just have additional heads trained to predict each SOFA's input feature, and successively apply CTSketch on top of it to predict the score?
> >
> > - Thanks for the additional details on the computation of the equation in Lines 135-136. I suggested that the authors integrate these into the manuscript, as they clarify a lot about how the inference with sketches is done in practice.
> >
> > - From the additional results provided on the ablation of the two components (sketching and task decomposition), it seems clear that the combination of them is not always the best solution that one can target. Instead, it seems to me that if it is possible to decompose the task using full vectors, this should be the first measure to target, as it reduces the computation time without penalizing accuracy. Sketching, on the other hand, should only be used when the former is not/only partially possible. Is this interpretation correct?
> >
> > - Generalization to other programs: What you are saying is that only the neural component (i.e., the digit classifier) can be reused across programs?

---

> > > ### Author Response · Authors · 2025-08-04
> > >
> > > Thank you for the response and additional questions.
> > >
> > > > Being aware of the limitations that LLMs face, I am to some extent skeptical about their applicability to the program synthesis task, at least for tasks of complexity greater than Sum16 and Visudo. However, I am also aware that this goes beyond the scope of the paper, so I acknowledge that this does not represent a limitation of the current submission (but, rather, of the long-term applicability of the proposed technique).
> > >
> > > Thank you for bringing up this point, and we will definitely consider the limitations of LLMs when exploring synthesis techniques in future work.
> > >
> > > > Regarding the additional real-world experiment you mention on EHR, it is not exactly clear what the role of CTSketch is in this case. Would you just have additional heads trained to predict each SOFA's input feature, and successively apply CTSketch on top of it to predict the score?
> > >
> > > Your understanding is correct. A program that diagnoses sepsis from SOFA scores would take the features that can indicate the severity of organ dysfunction (e.g., respiratory rate, level of bilirubin, etc.). The program would score each of these features according to the SOFA scoring system, and return true if the features meet the sepsis-3 criteria (SOFA score greater than or equal to 2). The CTSketch tensor would represent the possible inputs (predicted EHR features) and outputs (yes/no sepsis) to this program. To clarify, we haven’t tried implementing this yet, but this is just one avenue we are considering for using CTSketch to make real-world predictions. However, we believe that our results on synthetic tasks (in terms of accuracy and efficiency) indicate that CTSketch would probably do well in these real-world settings.
> > >
> > > > Thanks for the additional details on the computation of the equation in Lines 135-136. I suggested that the authors integrate these into the manuscript, as they clarify a lot about how the inference with sketches is done in practice.
> > >
> > > We appreciate the feedback, and we will make sure to include this equation in the revised manuscript.
> > >
> > > > Instead, it seems to me that if it is possible to decompose the task using full vectors, this should be the first measure to target, as it reduces the computation time without penalizing accuracy. Sketching, on the other hand, should only be used when the former is not/only partially possible. Is this interpretation correct?
> > >
> > > Yes, this interpretation is correct. Using full-rank tensors yields better performance, while program decomposition can further reduce training time. However, even with decomposition, handling full-rank tensors can quickly become computationally infeasible. Tensor sketching efficiently approximates the program without significant loss in performance, enabling scalability to larger problems.
> > >
> > > > Generalization to other programs: What you are saying is that only the neural component (i.e., the digit classifier) can be reused across programs?
> > >
> > > Yes, we are saying that only the neural component can be reused across programs. While the full tensor summary cannot be reused across tasks, we want to point out that sub-programs often can be reused, e.g., if the sum4 task is decomposed and uses sum2 tensors in its first layer, then these sum2 tensors can also be used in a decomposed sum8 program. That being said, we want to reiterate that we found initializing program tensors to be extremely efficient, taking less than a second for each benchmark's program.

---

> > > > ### Comment · Reviewer_hw4L · 2025-08-05
> > > >
> > > > Thanks for the additional responses. Overall, the rebuttal contributed to clarify a lot of my concerns regarding the submission. Hence, I will raise my initial score accordingly.

---

### Note · Authors · 2025-08-13

We thank the reviewers for the valuable feedback, and we would like to summarize the points that we addressed in the rebuttal and how we will incorporate them in the final version.

## Decomposition vs. sketching

One common comment was that the benefits of program sketching vs. decomposition were not properly addressed. To clarify this point, we ran an ablation study that compares the performance gains from program decomposition and tensor sketching. With the same decomposed program structure, we observe that sketching offers substantial memory savings, albeit with some loss in accuracy. When scaling to higher-dimensional input spaces, memory becomes a critical bottleneck – program decomposition alone is often insufficient, which highlights the necessity of tensor sketching for scaling. Within full-rank, the gains from greater degree of decomposition are clear, both in terms of training time and memory efficiency. However, this trend reverses with sketches, due to the use of the RBF kernel at the output of each program layer.

## Program tensor initialization

One concern was that the program tensor initialization requires intermediate labels, which means that CTSketch is able to “cheat” on the tasks. We clarified that we do not require any labeled data for tensor initialization because we use structured program input/output pairs.

Additionally, there was a concern about the generalizability of the program tensor to other tasks, e.g., the sum4 program tensor cannot be used for sum5. But this, to the best of our knowledge, is true for any neurosymbolic framework. For example, the DeepProbLog program for sum4 could not be used to train the sum5 task.

## Revisions

In the revised manuscript, we plan to include the ablation on decomposition vs. sketching. Additionally, we will add a section to the related work on arithmetic/probabilistic circuits and how they relate to CTSketch. We will include the references suggested by the reviewers. We will also make it clear that we are introducing program decomposition with tensor summaries, not just program decomposition (which has been used in prior works). Lastly, we will address all unclear statements and grammatical errors pointed out by the reviewers.

---

### Decision · Program_Chairs · 2025-09-17

**Decision:**

Accept (poster)

**Comment:**

This submission introduces a compositional tensor sketching method designed to scale neurosymbolic learning. The reviewers generally agreed that the contribution is technically solid, with experiments demonstrating both efficiency and accuracy. Some concerns were raised about the breadth of applications and experimentation, but the rebuttal largely addressed the concerns. Overall, the committee reached consensus that this contribution advances the scalability of neurosymbolic methods and will be of interest to the NeurIPS community. I recommend acceptance.